# REFERENCE-BASED CATEGORY DISCOVERY: UNSUPERVISED OBJECT DETECTION WITH CATEGORY AWARENESS

## ABSTRACT

Existing unsupervised methods fail to generate category labels or learn category-aware features. In contrast, one-shot detectors depend on manual category labels to perform sample assignment and classification loss calculation. As a result, these detectors are not suitable for unsupervised scenarios where no category labels are available. To overcome these limitations, we propose **Ref**erence-based **C**ategory **D**iscovery (RefCD), an unsupervised detector that enables category-aware[1] detection without any manually annotated labels. Specifically, It leverages feature similarity between predicted objects and unlabeled reference images. Unlike previous unsupervised methods that lack category awareness and one-shot methods which require labeled data, RefCD introduces a carefully designed Feature Similarity (FS) loss, which explicitly guides the learning of potential category-aware features, enabling RefCD to acquire category-aware information in an unsupervised paradigm without category labels. Additionally, RefCD supports category-agnostic detection without reference images, serving as a unified framework. Comprehensive quantitative and qualitative analysis of category-aware and category-agnostic detection results demonstrates its effectiveness.

## 1 INTRODUCTION

Object detection aims to localize and categorize objects within images. Traditional methods heavily rely on manually annotated datasets Lin (2014); Shao & Li (2019), which are not only time-consuming and labor-intensive but also suffer from poor generalization to unseen categories. Moreover, these methods are confined to predefined categories, inherently limiting their scalability.

To reduce the reliance of object detection on manual annotations, unsupervised object detection has been proposed. Non-deep learning object detection method Vo et al. (2021) uses graph-based algorithms to detect objects. Early deep learning based studies Wang et al. (2022); Bar et al. (2022) adopt complex architectures for detector training and generate coarse pseudo-boxes. Recent methods Wang et al. (2023b;a) explore purely unsupervised learning for object detection, where self-supervised Vision Transformers (ViT) are used to generate pseudo-boxes for detector training. Specifically, TokenCut Wang et al. (2023b) and CutLER Wang et al. (2023a) utilize attention maps from DINO Caron et al. (2021) to perform normalized cuts for pseudo-box generation. Despite these advances, all of these unsupervised methods are confined to category-agnostic detection and lack the capability of object classification, as illustrated in Figure 1 (b).

To improve generalization of unseen categories, one-shot detection Yan et al. (2019); Han et al. (2024); Liu et al. (2024a;b) and open-vocabulary detection Zang et al. (2022); liu (2024); Minderer et al. (2022); Zhong et al. (2022) have been proposed to address the limitations of closed-set by identifying objects beyond predefined categories. Open-vocabulary methods leverage pre-trained vision-language models Devlin et al. (2019); Radford et al. (2021), they are constrained by a large number of model parameters and fine-grained annotations, resulting in substantial labor and computational costs. However, one-shot detection methods also require fine-grained annotated data and

---

[1] Unlike traditional methods that detect objects with pre-defined category labels, this work detects objects that share the same category with the unlabeled reference image provided on-the-fly.

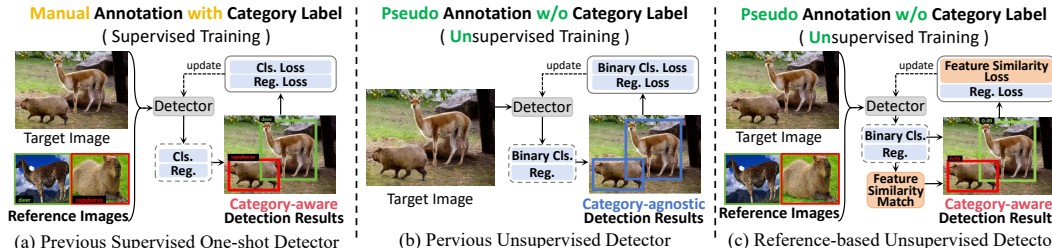

Figure 1: Comparison of different detection paradigms. (a) Previous one-shot detection methods Yan et al. (2019); Han et al. (2024); Liu et al. (2024b;a) rely on manual annotations for supervised training and one-shot fine-tuning to enable detectors to learn novel categories. (b) Previous unsupervised methods Wang et al. (2023b;a) employ unsupervised models to generate pseudo boxes for detector training. They localize all foreground objects without category awareness. (c) The proposed unsupervised method introduces reference images and feature similarity loss, constraining the model to identify similar objects through feature similarity without any category label. (Cls. and Reg. denote Classification and Regression, respectively.)

even one-shot fine-tuning (Figure 1(a)). They neither reduce annotation costs nor avoid the requirement for high annotation accuracy. In contrast, our work focuses on unsupervised object detection, aiming to completely eliminate reliance on manual annotations, which is a key drawback that existing one-shot methods fail to overcome.

To bridge this gap, this paper proposes an unsupervised **Ref**erence-based **C**ategory **D**iscovery (RefCD) detector. As shown in Figure 1(c), RefCD enables category-aware object detection without any manual annotations. Unlike one-shot methods that require labeled data or unsupervised methods that ignore category information, RefCD leverages feature similarity between predicted objects and unlabeled reference images to guide the learning of category-aware features. Specifically, it uses reference images to introduce contextual semantics and proposes a Feature Similarity (FS) loss function. This loss is pivotal as it differs fundamentally from supervised loss functions, which rely on explicit category labels. In contrast, FS loss operates in an unsupervised manner, guiding the detector to learn potential category-specific features by maximizing similarity between objects of the same implicit category and minimizing it for others.

The proposed RefCD aims to locate all objects via category-agnostic detection and then identify specific category objects through category-aware object matching. In the category-aware object matching, we match the category-agnostic detection results with the reference image based on their feature similarities. During training, an image patch cropped from the target image with a randomly selected pseudo box is used as a reference image to guide the detector to locate the object within the selected pseudo box. Other objects that share high feature similarities with the reference image are also identified. In the absence of category labels, the novel FS loss explicitly guides the detector to learn potential category-specific features. As a result, RefCD can infer category information from features without relying on explicit category labels. Additionally, RefCD can perform category-agnostic object detection, a task enhanced by the category-aware feature learning. Our main contributions are as follows:

- We propose an unsupervised reference-based detection method that enables open-set object detection with reference images without requiring any annotations.

- We introduce a feature similarity loss that encourages the detector to mine the potential category information of different objects during unsupervised training.

- Experimental results show that RefCD achieves state-of-the-art performance in unsupervised object detection, even competitive with some supervised detectors.

## 2 RELATED WORK

This section briefly reviews related works for object detection from different aspects, including unsupervised object detection and one-shot object detection.

## 2.1 One-shot Object Detection

One-shot object detection methods are proposed to reduce the annotation cost of novel categories and break the limitations of pre-defined closed-set categories. Traditional training-based methods Yan et al. (2019); Zhang et al. (2022a); Han et al. (2024) pre-train on base categories and then fine-tune with one-shot samples from novel categories. SimTrans Chen et al. (2021) applies similarity transfer to the learning of novel categories. SimFormer Chen et al. (2022) proposes a complementary loss to explore the positions of novel categories based on the pixels of base categories. But the lack of strong supervision for novel classes affects the performance of detectors when handling novel categories. Recent methods have introduced reference-based object detection Osokin et al. (2020); Zang et al. (2022); Li et al. (2022); Liu et al. (2024a;b), where flexible reference images serve as prompts during inference, replacing additional one-shot fine-tuning. Early methods Huang et al. (2020) leverage single object tracking methods to detect all objects in the image that share the same category as the reference image. More recently, Siamese-DETR Liu et al. (2024a) integrates reference image features as queries into a Transformer-based framework to detect objects of the same category, while SINE Liu et al. (2024b) introduces an in-context interaction module and ID queries to enhance model accuracy and flexibility. However, the aforementioned methods also rely on labor-intensive and fine-grained category annotations to help the model accurately associate references with target objects. In contrast, the proposed method eliminates expensive annotation costs and effectively detects objects of interest categories through unsupervised training.

## 2.2 Unsupervised Object Detection

Unsupervised object detection Xie et al. (2021); Bar et al. (2022); Siméoni et al. (2021); Wang et al. (2022; 2023a;b) aims to eliminate the reliance on annotated data by unsupervised training. Traditional methods employ unsupervised generative-based Donahue et al. (2016); Donahue & Simonyan (2019); Brock (2018) and contrastive learning Goodfellow et al. (2014); Xie et al. (2021) methods to enhance image feature representations, followed by supervised fine-tuning to reduce annotation costs during training. DINO Caron et al. (2021) has been proven effective in extracting salient object features from images. Based on this observation, LOST Siméoni et al. (2021) and TokenCut Wang et al. (2023b) leverage self-supervised ViT to extract patch-based feature maps, segmenting the most prominent object from each image. MaskDistill Gansbeke (2022) generates initial object masks from the affinity maps generated by DINO. FreeSOLO Wang et al. (2022) employs the FreeMask stage to generate multiple coarse masks for each image, refining them through self-training to achieve unsupervised object detection. CutLER Wang et al. (2023a) introduces a foreground attention mask after each NCut step, generating multiple pseudo boxes in a single image. Recently, U2Seg Niu et al. (2024) clusters the pseudo box features to generate pseudo category labels and matches the pseudo labels with the corresponding category names during inference.

However, most of these methods focus on generating pseudo boxes/masks, limiting the trained detectors to object localization rather than object classification. This work leverages reference images to train the detector for feature similarity matching. Mining category information from object features in the absence of category labels while being free from the constraints of pre-defined categories.

## 3 Method

This section introduces motivation and overview of RefCD, provides a detailed explanation of the feature similarity loss, and finally presents its training strategy.

### 3.1 Motivation

Supervised learning-based detectors Ren et al. (2016); Carion et al. (2020) depend on manual annotations. Although unsupervised methods Wang et al. (2023a;b) reduce the dependence on manual annotations, they cannot generate pseudo boxes with category labels. This makes it impossible for the detector to acquire classification capabilities through unsupervised learning. At the same time, traditional classification losses (*e.g.*, cross entropy loss Mao et al. (2023), focal loss Lin et al. (2017)) limit the detector to predefined categories, thereby restricting its processing of novel categories. To solve this problem, we propose the RefCD. This method uses the reference image as a category

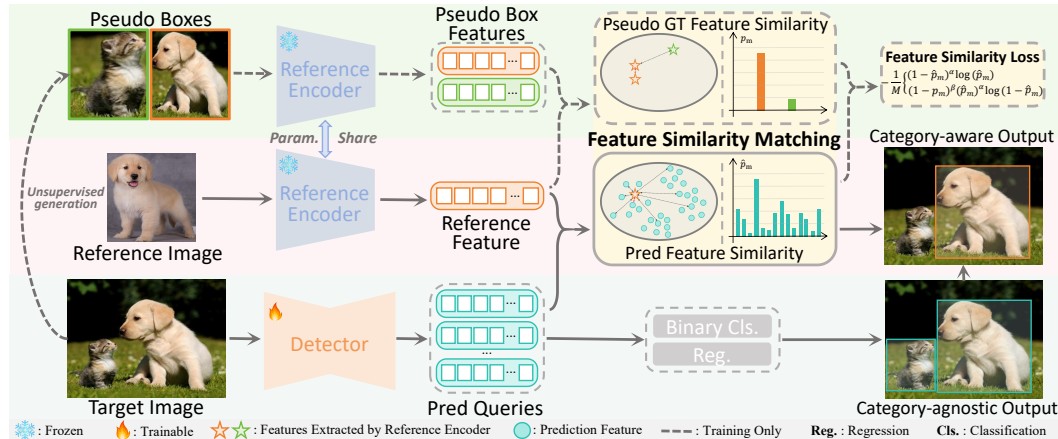

Figure 2: **Overview of RefCD.** The reference image features are used as category prompts for detecting objects of interest, with the predicted object determined by the similarity between features. Reference features and pseudo box features are extracted by a frozen reference encoder Oquab et al. (2023).The detector is trained with traditional object detection losses and the proposed feature similarity loss, enabling both category-agnostic object detection and category-aware object matching

prompt and replaces label predictions with feature similarity matching. This enables the detector to acquire open-set detection capabilities through unsupervised learning.

## 3.2 OVERVIEW OF THE PROPOSED REFCD

The overview of the proposed RefCD is shown in Figure 2. It mainly consists of two components: the reference encoder and the detector. The reference encoder is a self-supervised pre-trained ViT Oquab et al. (2023), and the detector is a transformer-based detector Zhao et al. (2024).

In the training stage, the reference encoder is frozen, and only the detector is trainable. Following previous methods, pseudo boxes are generated for unsupervised training. To achieve category-aware detection, RefCD additionally trains on category-aware object matching, and the feature similarity loss is used to calculate the matching loss. In the inference stage, RefCD does not perform explicit classification. It detects objects of interest based on reference images provided by users. To conduct a quantitative evaluation, grounded categories are defined by users during inference. Please note that category definitions are not involved during either training or inference, as RefCD focus on the learning of feature similarity between objects and reference images.

In the category-agnostic object detection, all objects are treated as foreground and classified by a binary classification head. The commonly used Hungarian loss Carion et al. (2020) is used to optimize this task. For category-aware object matching, the reference image is used to match the category-agnostic detection results to produce category-aware detection results. Since the category of pseudo boxes and the reference image are absent, we propose a new Feature Similarity (FS) loss to optimize this task. In the following, we first elaborate on the FS loss and then introduce the training details of the combination of the two tasks.

## 3.3 FEATURE SIMILARITY LOSS

Given a target image, we first generate the pseudo boxes using existing unsupervised work Wang et al. (2023a) with the help of the reference encoder. For each pseudo box, the corresponding image patch is cropped and fed into the reference encoder. The outputted class token is used as the feature representation of the pseudo box. Formally, we denote the pseudo annotations as $A = \{a_m | a_m = (\boldsymbol{b}_m, \boldsymbol{f}_m), m = 1, \cdots, M\}$, where $M$ is the number of pseudo boxes, $\boldsymbol{b}_m \in \mathbb{R}^4$ is the $m$-th bounding box, and $\boldsymbol{f}_m \in \mathbb{R}^D$ is the feature representation for the $m$-th pseudo box with $D$ as the dimensionality of reference encoder. As for the reference image, we randomly sample an index $m_{ref}$ from $\{0, ..., M\}$, and use $\boldsymbol{f}_{m_{ref}}$ as the feature of the reference image.

By feeding the target image into the detector, we can get the category-agnostic detection results, which are denoted as $\hat{Q} = \{\hat{q}_n | \hat{q}_n = (\hat{\boldsymbol{q}}_{c_n}, \hat{\boldsymbol{q}}_{b_n}), n = 1, \cdots, N\}$, where $N$ is the number of predictions, $(\hat{\boldsymbol{q}}_{c_n}, \hat{\boldsymbol{q}}_{b_n})$ is the $n$-th outputted query with $\hat{\boldsymbol{q}}_{c_n} \in \mathbb{R}^D$ as the query content and $\hat{\boldsymbol{q}}_{b_n} \in \mathbb{R}^4$ as the query box (*i.e.*, the predicted box).

The FS loss is designed to help the detector identify the objects that share the same category as the reference image. In the training stage, FS loss involves two steps: assigning the pseudo annotations to the detection results and calculating the loss between assignment results.

**Assignment of pseudo annotations.** Traditional DETR series detectors need to assign the annotations to detection results based on category labels and boxes, where each annotation is matched with a prediction for loss computation. Let $I = \{(m_n, n) | m_n = 1, \cdots, M\}$ be the matched index pairs of the assignment, where $(m_n, n)$ represents that the $m_n$-th box is match with the $n$-th query. The matching procedure is finished by the Hungarian algorithm

$$I = \text{Hungarian}(-\text{Cost}(A, \hat{Q})), \tag{1}$$

where Hungarian$(\cdot)$ refers to the Hungarian algorithm that returns the matching result with the minimum total cost based on the cost matrix. Cost$(\cdot, \cdot)$ represents the function that calculates the cost matrix for all annotations and all predictions. Specifically, the cost between the annotation $a_m$ and the prediction $\hat{q}_n$ in existing supervised DETR series detectors is calculated as

$$\text{Cost}(a_m, \hat{q}_n) = \mathcal{C}_{\text{cat}}(a_m, \hat{q}_n) - \mathcal{D}_{\text{box}}(\boldsymbol{b}_m, \hat{\boldsymbol{q}}_{b_n}), \tag{2}$$

where $\mathcal{C}_{\text{cat}}(\cdot, \cdot)$ returns the predicted confidence of the query that corresponds to the annotated category of $a_m$ Carion et al. (2020), while $\mathcal{D}_{\text{box}}(\cdot, \cdot)$ calculates the distance between the provided two boxes based on the smooth L1 loss and GIoU loss. A smaller distance means that the two boxes are closer to each other Carion et al. (2020).

In the unsupervised case, the category labels are unable to be produced for pseudo boxes, indicating that the confidence for a specific category is inaccessible, which hinders the assignment based on the cost in Eq.(2). To address this, we replace the predicted confidence with feature similarity as the category information for the assignment. In detail, the feature similarity between $a_m$ and $\hat{q}_n$ is calculated by a cosine-based similarity between $\hat{\boldsymbol{q}}_{c_n}$ and $\boldsymbol{f}_m$, and the final cost between $a_m$ and $\hat{q}_n$ for the assignment in our unsupervised detector is calculated as

$$\text{Cost}(a_m, \hat{q}_n) = \mathcal{S}_{\cos}(\boldsymbol{f}_m, \hat{\boldsymbol{q}}_{c_n}) - \mathcal{D}_{\text{box}}(\boldsymbol{b}_m, \hat{\boldsymbol{q}}_{b_n}), \tag{3}$$

Where $\mathcal{S}_{\cos}(\cdot, \cdot)$ calculates the cosine similarity of the two given feature vectors and then activates it by the Sigmoid function with temperature 10. With the newly designed cost in Eq.(3), the matched index pairs can be obtained through Eq.(1) with the absence of category labels.

**Loss calculation.** In the assignment step, the feature similarity between predictions and pseudo boxes is used as the cost. It requires that the content of queries and the feature representation of pseudo boxes are in the same latent space, and the same category objects share a higher feature similarity. Intuitively, for a matched query, its content should be similar to the feature representation of its matched pseudo box. For an unmatched query, its content should be dissimilar of all pseudo boxes. Let $I_Q$ be the index of matched queries in $I$. The content of queries and the feature representation of pseudo boxes can be constrained by the cosine-based similarity

$$\mathcal{L}_{cos} = \frac{1}{N} \cdot \sum_{n=1}^{N} \begin{cases} -\mathcal{S}_{\cos}(\boldsymbol{f}_{m_n}, \hat{\boldsymbol{q}}_{c_n}), & \text{if } n \in I_Q, \\ \max(\{\mathcal{S}_{\cos}(\boldsymbol{f}_m, \hat{\boldsymbol{q}}_{c_n}) | m = 1, ..., M\}), & \text{else.} \end{cases} \tag{4}$$

However, we find that the loss in Eq.(4) results in inferior category-aware detection results. The reason is that the category-aware detection results are matched and selected based on the feature similarity between the query content and the feature representation of reference images. But the feature representation of the reference image is not used in Eq.(4), resulting in a gap between the training and inference stages. Inspired by the previous work Zhou et al. (2019), we propose to calculate the loss of the feature similarity

$$\mathcal{L}_{FS} = \frac{-1}{N} \cdot \sum_{n=1}^{N} \begin{cases} (1 - \hat{p}_n)^\alpha \log(\hat{p}_n), & \text{if } n \in I_Q \text{ and } m_n = m_{ref}, \\ (1 - p_n)^\beta (\hat{p}_n)^\alpha \log(1 - \hat{p}_n), & \text{else.} \end{cases} \tag{5}$$

where $\alpha = 2$ and $\beta = 4$ are hyperparameters, $\hat{p}_n$ is the similarity between the $n$-th query and the reference image, and $p_n$ is the pseudo similarity used to supervise $\hat{p}_n$, which are defined as

$$\hat{p}_n = \mathcal{S}_{\cos}(\boldsymbol{f}_{m_{ref}}, \hat{\boldsymbol{q}}_{c_n}), \tag{6}$$

$$p_n = \begin{cases} \mathcal{S}_{\cos}(\boldsymbol{f}_{m_{ref}}, \boldsymbol{f}_{m_n}), & \text{if } n \in I_Q \text{ and } \mathcal{S}_{\cos}(\boldsymbol{f}_{m_{ref}}, \boldsymbol{f}_{m_n}) \geq \tau, \\ 0, & \text{else,} \end{cases} \tag{7}$$

where $\tau = 0$ is the threshold for selecting positive samples. The intuition behind the design of $\mathcal{L}_{FS}$ is that for a matched pair, the similarity between the query and the reference image should be the same as the similarity between the pseudo box and the reference image. The feature similarity loss is beneficial to the inference stage, where the category-aware detection results are matched and selected based on the similarity between queries and the reference image. In addition, while dynamically changing the reference image in the inference stage, RefCD can perform open-set object detection.

## 3.4 TRAINING OF REFCD

The proposed RefCD is trained for the category-aware object detection task. The loss function is

$$\mathcal{L} = \mathcal{L}_{hung} + \lambda \mathcal{L}_{FS}, \tag{8}$$

where $\mathcal{L}_{hung}$ is the commonly used Hungarian loss Carion et al. (2020) in existing supervised DETR series detectors for category-agnostic objects detection in this work, and $\lambda$ is the weight of feature similarity loss. During training, the number of queries far exceeds the number of pseudo boxes, resulting in most queries being unable to match with pseudo boxes. Therefore, we only use the top-K queries with the highest *foreground* confidence to calculate the feature similarity loss.

## 4 EXPERIMENTS

In this section, we first introduce datasets and implementation details. Then, we present the results of RefCD versus existing methods in category-aware and category-agnostic object detection. Next, we provide a discussion to show the effectiveness of our designs. Finally, we showcase the scalability of RefCD in single object tracking.

## 4.1 DATASET AND SETTINGS

Following previous work Wang et al. (2023a), we generate pseudo boxes for the images in **ImageNet** Deng et al. (2009) dataset (about 1.3M images). We conduct a detailed evaluation on the widely used COCO Lin (2014) and GMOT-40 Bai & Cheng (2021) datasets. Without specification, our RefCD is only trained with pseudo boxes on ImageNet.

**COCO** is a standard object detection benchmark with 118K images across 80 categories. We evaluate category-agnostic detection on COCO 20K and COCO val2017. For open-set one-shot evaluation, following Liu et al. (2024b), we split COCO into COCO NOVEL (20 categories overlapping with PASCAL VOC Everingham et al. (2010)) for evaluation.

**GMOT-40** consists of 40 video scenes (10 categories) with 9.6K images. Only same-category objects are annotated per scene. All images are used for one-shot evaluation.

**Implement details.** The ViT-large model pre-trained with DINOv2 is used as the reference encoder and the detector is RT-DETR Zhao et al. (2024) with ResNet50 He et al. (2016) pre-trained with unsuerpvised method ReLiCV2 Tomasev et al. (2022) as the backbone. Following the CutLER Wang et al. (2023a), we generate pseudo boxes on ImageNet Deng et al. (2009) and refine them through self-training, more details are shown in Section A.6.1. The model is trained on 2 NVIDIA RTX 3090 GPUs for 80K iterations with a batch size of 16, and an exponential moving average (EMA) with a decay rate of 0.9999 is applied. Moreover, during each training iteration, four images are randomly selected and tiled to form a new larger target image with more pseudo boxes.

At the training stage, all (manual, more specifically) annotations of RefCD can be eliminated. During the inference stage, reference images are typically provided by users. While no explicit category labels are attached to these reference images, but category concepts are implicitly grounded within them by users. Notably, the grounded class categories of reference images are defined solely for quantitative evaluation, which facilitates unified metric-based assessment.

Table 1: **One-shot detection results (category-aware)** on COCO NOVEL and GMOT-40. We report learning paradigms and pretraining data. Pre-defined categories methods train detectors on pre-defined category data and undergo one-shot fine-tuning with novel categories. Reference-based methods use novel category objects as reference images during inference. Some existing methods' results on COCO NOVEL are accessed from papers as some are not publicly available.

| Method | Publication | Paradigm | Pretrain | COCO NOVEL | | | GMOT-40 | | |
|---|---|---|---|---|---|---|---|---|---|
| | | | | $AP$ | $AP_{50}$ | $AR$ | $AP$ | $AP_{50}$ | $AR$ |
| *pre-defined categories methods* | | | | | | | | | |
| Meta R-CNN Yan et al. (2019) | ICCV'19 | Supervised | COCO BASE | 1.5 | - | - | 6.1 | 10.3 | 14.9 |
| iMTFA Ganea et al. (2021) | CVPR'21 | Supervised | COCO BASE | 3.3 | 6.0 | - | - | - | - |
| iFS R-CNN Nguyen & Todorovic (2022) | CVPR'22 | Supervised | COCO BASE | 6.4 | - | - | - | - | - |
| Meta-DETR Zhang et al. (2022a) | TPAMI'22 | Supervised | COCO BASE | 7.5 | 12.5 | - | 10.4 | 22.1 | 26.4 |
| RefT Han et al. (2024) | TPAMI'24 | Supervised | COCO BASE | 5.2 | - | - | - | - | - |
| *reference-based methods* | | | | | | | | | |
| GlobalTrack Huang et al. (2020) | AAAI'20 | Supervised | LaSOT+GOT-10K+COCO | - | - | - | 9.6 | 28.3 | 18.3 |
| SINE Liu et al. (2024b) | NeurIPS'24 | Supervised | COCO BASE | 10.5 | - | - | - | - | - |
| UNICL-SAM Sheng et al. (2025) | CVPR'25 | Supervised | COCO BASE | 13.2 | - | - | - | - | - |
| RefCD (Ours) | | Unsupervised | ImageNet | **15.1** | **24.7** | **29.3** | 12.9 | **31.8** | **30.2** |

Figure 3: Qualitative results of RefCD on COCO. Reference image are shown on the left side of the dashed line, and the one-shot detection results are displayed on the right side. Please note that to facilitate understanding, we add category names to the detection boxes. In actual inference, it is unnecessary to predefine categories for reference images.

## 4.2 CATEGORY-AWARE EVALUATIONS

We compare our RefCD with existing supervised one-shot detection methods: one-shot fine-tuning methods Meta R-CNN, MTFA, iFS R-CNN, Meta-DETR, RefT; and reference-based methods GlobalTrack, SINE, UNICL-SAM. Following previous methods, we report results using 4 randomly selected groups of reference images. Detailed ablation experiments are presented below.

**Comparison on COCO NOVEL.** As shown in Table 1, we compare the proposed method with various one-shot detectors. As expected, the limitations of pre-defined categories methods become apparent as they struggle to learn the features of novel categories during fine-tuning with a single image. This is particularly evident in Meta R-CNN and MTFA, where the Region Proposal Network (RPN) often identifies objects from novel categories as background, leading to a poor one-shot detection performance. In contrast, reference-based detectors focus on learning the similarity between object features, reducing the constraints imposed by category labels, and demonstrating more robust detection performance on novel categories. Compared to the existing state-of-the-art supervised method UNICL-SAM, our RefCD achieves a 1.9% AP improvement on COCO NOVEL. We show qualitative results of RefCD on COCO in Figure 3.

**Comparison on GMOT-40.** Table 1 presents a comparison of detection results on GMOT-40. GMOT-40 consists of 40 video sequences, where we randomly crop an object from the first frame of each video as the reference image. Due to the high similarity among objects, one-shot methods demonstrate better performance on GMOT-40. But the domain gap between training and evaluation data still limits the performance of pre-defined category methods. GlobalTrack is originally designed for single object tracking task. Although its robust global search mechanism allows it to be extended to one-shot detection, it is difficult to distinguish different objects of the same category. The proposed method addresses the challenges of distinguishing different objects and handling cross-domain variations, achieving 31.8% $AP_{50}$ and 30.2% $AR$ on GMOT-40.

## 4.3 CATEGORY-AGNOSTIC EVALUATIONS

In the category-agnostic object detection, all objects are treated as foreground and classified by a binary classification. We compare our RefCD with existing unsupervised training methods, including

Table 2: **Unsupervised object detection results (category-agnostic)** on COCO 20K and COCO val2017. We report detection metrics along with pretraining data and backbone networks. Semi-supervised methods in the top half of the table are pre-trained on ImageNet and fine-tuned on COCO, while the unsupervised methods below are trained solely on ImageNet.

| Method | Publication | Backbone | Pretrain | COCO 20K | | | COCO val2017 | | |
|---|---|---|---|---|---|---|---|---|---|
| | | | | $AP$ | $AP_{50}$ | $AR$ | $AP$ | $AP_{50}$ | $AR$ |
| *Semi-supervised methods* | | | | | | | | | |
| LOST Siméoni et al. (2021) | arXiv'21 | DINO | ImageNet+COCO | 1.1 | 2.4 | - | - | - | - |
| MaskDistill Gansbeke (2022) | arXiv'22 | MoCo | ImageNet+COCO | 2.9 | 6.8 | - | - | - | - |
| FreeSOLO Wang et al. (2022) | CVPR'22 | DenseCL | ImageNet+COCO | 4.1 | 9.7 | - | 4.2 | 9.6 | - |
| *unsupervised methods* | | | | | | | | | |
| DETReg Bar et al. (2022) | CVPR'22 | SwAV | ImageNet | - | - | - | 1.0 | 3.1 | - |
| DINO Zhang et al. (2022b) | arXiv'22 | DINO | ImageNet | 0.3 | 1.7 | - | - | - | - |
| TokenCut Wang et al. (2023b) | TPAMI'23 | DINO | ImageNet | - | - | - | 3.0 | 5.8 | 8.1 |
| CutLER Wang et al. (2023a) | CVPR'23 | DINO | ImageNet | 10.1 | 21.8 | 30.0 | 10.2 | 21.3 | 29.6 |
| RT-DETR Zhao et al. (2024) | CVPR'24 | ResNet50 | ImageNet | 12.8 | 21.7 | 29.3 | 11.8 | 21.4 | 28.4 |
| CuVLER Arica et al. (2024) | CVPR'24 | ResNet50 | ImageNet | 13.1 | 24.1 | - | 12.8 | 23.5 | - |
| RefCD (Ours) | | ResNet50 | ImageNet | **13.3** | **24.5** | **34.8** | **12.9** | **23.6** | **33.9** |

LOST, MaskDistill, FreeSOLO, DETReg, DINO, TokenCut, CutLER, and RT-DETR. Category-agnostic object detection results are shown in Table 2. By removing the feature similarity matching and the reference encoder, the proposed method produces category-agnostic detection results. It can be observed that the proposed method outperforms CutLER by 3.0% AP and 2.8% AR on COCO 20K and 2.7% AP and 2.8% AR on COCO val2017 using the same pseudo boxes generation method, and reaches comparable performance as CuVLER. This demonstrates that RefCD exhibits effective performance in the category-agnostic detection task. Specifically, feature similarity based sample assignment provides higher-quality positive samples for detector training, while features containing deeper semantic information contribute to more accurate bounding box predictions. This further demonstrates the advantages of unsupervised learning, where the detector can surpass the limitations of manual annotations and identify previously undefined objects.

## 4.4 DISCUSSION

In the following, we provide discussions to show the effectiveness of our method and report both category-agnostic and category-aware detection results on COCO val2017 and COCO NOVEL.

**Importance of feature similarity loss.** Table 3 demonstrates the importance of feature similarity loss and its contribution to detection performance. Experimental results show that constraining query content supervision with pseudo box feature representations is important. Without this constraint, the detector fails to learn category information, leading to irregular and random similarity between predicted queries and pseudo box features, making category-aware object matching infeasible. Compared to a simple cosine similarity constraint $\mathcal{L}_{cos}$, $\mathcal{L}_{FS}$ addresses the gap between the training and inference stages. Training RefCD with $\mathcal{L}_{FS}$ achieves 2.4% AP and 9.1% AR higher than training with $\mathcal{L}_{cos}$ in category-aware detection. Additionally, it can be observed that category-aware feature learning implicitly enhances the category-agnostic detection performance. The best result of RefCD achieves 0.7% AP and 4.0% AR higher than training with $\mathcal{L}_{hung}$.

Table 3: Results of different loss.

| Loss | COCO val2017 (category-agnositic) | | | COCO NOVEL (category-aware) | | |
|---|---|---|---|---|---|---|
| | $AP$ | $AP_{50}$ | $AR$ | $AP$ | $AP_{50}$ | $AR$ |
| $\mathcal{L}_{hung}$ | 11.8 | 21.4 | 28.4 | 0.1 | 0.3 | 5.1 |
| w/ $\mathcal{L}_{cos}$ | 12.1 | 21.8 | 30.0 | 12.6 | 20.1 | 19.6 |
| w/ $\mathcal{L}_{FS}^{K=300}$ | 12.5 | 23.2 | 32.8 | 15.0 | 24.3 | 28.7 |
| w/ $\mathcal{L}_{FS}^{K=200}$ | 12.8 | 23.4 | 33.3 | 15.0 | 24.4 | 29.0 |
| w/ $\mathcal{L}_{FS}^{K=100}$ | **12.9** | **23.6** | **33.9** | **15.1** | **24.7** | **29.3** |
| w/ $\mathcal{L}_{FS}^{K=50}$ | 12.4 | 23.2 | 33.0 | 14.9 | 24.4 | 29.1 |

Table 4: Results of different feature similarity calculation methods

| Similarity Score | COCO NOVEL | | |
|---|---|---|---|
| | $AP$ | $AP_{50}$ | $AR$ |
| $S_{linear}(\cdot,\cdot)$ | 13.2 | 21.3 | 23.5 |
| $S_{euc}(\cdot,\cdot)$ | 14.6 | 23.5 | 26.9 |
| $S_{cos}(\cdot,\cdot)$ | **15.1** | **24.7** | **29.3** |

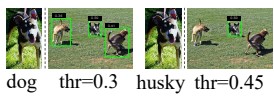

dog thr=0.3 husky thr=0.45

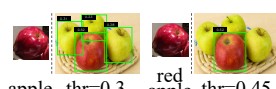

apple thr=0.3 red apple thr=0.45

Figure 4: Fine-grained grounding visualization.

**Discussion on feature similarity calculation.** As described in the Method section, we use $S_{cos}$ to calculate the similarity between predicted queries and pseudo box features. As shown in Table 4, we further discuss the impact of different feature similarity calculation methods on detection performance. Initially, we train RefCD using a learnable similarity prediction head, where features are fed into a linear layer to predict the similarity score $S_{linear}$ between objects and reference images. The results (1-st row of Table 4) indicate that the learnable linear layer tends to overfit the training data domain, leading to poor detection performance. Therefore, we employ Euclidean- $S_{euc}$ and Cosine

distance $S_{cos}$ to calculate similarity. Since the Euclidean distance has a range of $[0, +\infty)$, we apply an exponential function for activation. Compared to $S_{euc}$, $S_{cos}$ is more sensitive to categorical information in features. Consequently, when using $S_{cos}$, RefCD achieves the best performance.

**Discussion on reference image representations.** Table 6 presents the discussion results on reference image representations. During the training stage, RefCD uses a frozen reference encoder for pseudo boxes feature extraction. The extracted feature is used for the feature similarity matching and loss calculation (More details in Section A.6.2). To discuss the impact of different reference encoders on RefCD, we conduct experiments using unsupervised training ResNet Tomasev et al. (2022) and self-supervised training ViT Oquab et al. (2023). Notably, we use the deepest layer features of ResNet with average pooling, and similarly apply average pooling to ViT patch tokens.

As observed from the first three rows of Table 6, the pseudo boxes features extracted by ViT contain more category semantic information, and the accuracy increases as the hidden dimensions grow. Additionally, ViT outputs a separate class token, which contains more category information, helping the model learn more category features. Interestingly, the class tokens with more semantic information further improve the performance of category-agnostic detection. This suggests that better category feature information also contributes to more accurate bounding box regression.

**Discussion on different reference images.** We use four sets of reference images randomly selected from COCO training data and ImageNet for one-shot evaluation. As shown in Table 5, RefCD achieves stable results in both in-domain and cross-domain scenarios. Figure 3 shows that reference images of different categories can be used to distinguish objects of different categories. Different reference images of the same category produce different detection confidences, but the variation in detection results remains within an acceptable range, demonstrating the generalization capability of RefCD across different reference images. Although RefCD focuses on evaluating different categories, fine-grained category groups can be classified by adjusting the similarity threshold. As shown in Figure 4, higher confidence allows the RefCD to distinguish dog breeds and apple colors. Note that if multiple reference images are provided simultaneously, they are treated as distinct implicit categories. Detected objects are mutually exclusively assigned to the reference image with the highest feature similarity.

**Discussion on hyperparameter $K$ of feature similarity loss.** To mitigate the impact of negative samples on RefCD, we only use the top-$K$ queries with the highest foreground confidence for calculating the feature similarity loss. We perform ablation experiments on the hyperparameter $K$, and the results are presented in Table 3 (last 4 rows). When $K$ is set excessively large, the imbalance between positive and negative samples causes a decline in model performance. Specifically, RefCD achieves the optimal results when $K$ is set to 100.

**Discussion on hyperparameters $\alpha$ and $\beta$.** In the feature similarity loss, $\alpha$ balances the weights of positive and negative samples, and $\beta$ mainly controls the degree of suppression for simple samples. We conduct detailed ablation experiments on these two hyperparameters (Table 7), where $\alpha \in \{1, 2, 3, 4\}$ and $\beta \in \{3, 4, 5, 6, 7\}$. The best results are achieved when $\alpha = 2$ and $\beta = 4$.

Table 5: Results of different reference image sets.

| Ref. Sets | Ref. Source | COCO NOVEL | | |
|---|---|---|---|---|
| | | $AP$ | $AP_{50}$ | $AR$ |
| 1 | COCO | 14.6 | 24.3 | 29.0 |
| 2 | COCO | 15.3 | 25.1 | 29.4 |
| 3 | COCO | 15.7 | 25.7 | 29.7 |
| 4 | ImageNet | 14.7 | 24.1 | 28.8 |
| Avg. | - | 15.1 | 24.7 | 29.3 |

Table 6: Results of different reference image representations.

| Reference Representation | Hidden Dim. | COCO NOVEL | | |
|---|---|---|---|---|
| | | $AP$ | $AP_{50}$ | $AR$ |
| ResNet50 | 1024 | 11.3 | 18.2 | 21.8 |
| ViT-B (patch token) | 768 | 11.8 | 19.6 | 22.3 |
| ViT-L (patch token) | 1024 | 12.1 | 20.2 | 24.1 |
| ViT-B (class token) | 768 | 15.0 | 24.5 | 27.8 |
| ViT-L (class token) | 1024 | **15.1** | **24.7** | **29.3** |

Table 7: Results of FS loss with different $\alpha$ and $\beta$.

| $\alpha$ | $\beta$ | $AP_{50}$ | $AP$ | $AR$ |
|---|---|---|---|---|
| 1 | 3 | 23.5 | 14.7 | 28.2 |
| 1 | 4 | 22.1 | 12.9 | 23.3 |
| 2 | 4 | **24.7** | **15.1** | **29.3** |
| 2 | 5 | 24.0 | 14.9 | 27.7 |
| 3 | 5 | 23.8 | 15.0 | 28.9 |
| 3 | 6 | 23.7 | 14.7 | 28.7 |
| 4 | 6 | 23.5 | 14.7 | 28.8 |
| 4 | 7 | 22.3 | 13.9 | 28.0 |

**Discussion on tempearture parameter.** The temperature parameter serves to adjust the standard deviation between feature similarities. When temperature $\leq 1$, the feature similarity between positive and negative samples is very close, hindering the model from learning accurate category-aware features. When the temperature is excessively large (*e.g.*, temperature $\geq 100$), the predicted feature similarities tend to approach 0 or 1 after activation. This will lead to the neglect of some objects with high intra-category variation, reducing the generalization of RefCD. In category aware matching, we hope to have obvious differences in feature similarity between objects of different categories. An small temperature requires careful selection of a similarity threshold for each reference image.

The high temperature renders this threshold ineffective, as there are inevitably small feature differences between intra-class objects. Thus, when the temperature set to 10, RefCD can well adapt to a universal similarity threshold and achieve the optimal detection results (Table 8).

Table 8: Results of different temperature parameter.

| Temperature | $AP$ | $AP_{50}$ | $AR$ |
|---|---|---|---|
| 1 | 13.7 | 22.3 | 26.8 |
| 10 | 15.1 | 24.7 | 29.3 |
| 100 | 8.9 | 15.3 | 18.1 |

Table 9: Results of different $\tau$.

| $\tau$ | $AP$ | $AP_{50}$ | $AR$ |
|---|---|---|---|
| 0.9 | 13.4 | 8.0 | 15.7 |
| 0.6 | 18.7 | 11.4 | 20.4 |
| 0.3 | 22.8 | 13.2 | 27.1 |
| 0 | **24.7** | **15.1** | **29.3** |

Table 10: Results of weakly supervised training RefCD on COCO BASE.

| Method | Paradigm | Feature Level | $AP$ |
|---|---|---|---|
| RefCD | Weakly | Box | 24.9 |
| (Ours) | Supervised | Mask | 28.3 |

**Discussion on hyperparameter $\tau$ of feature similarity loss.** We explore the threshold $\tau$ for $p_n$ to filter out pseudo boxes that have low similarity with reference images, enabling RefCD to learn them as negative samples. As shown in Table 9, RefCD achieves the best category-aware detection results when $\tau = 0$ (*i.e.*, regressing $\hat{p}_n$ to $p_n$). We attribute this phenomenon to the potential intra-category variation. Since there are no accurate category labels, if $\tau > 0$ may exclude potential objects of interest at the initial training stage, resulting in an incorrect optimization direction. This specifically impairs generalization ability and open-set detection performance of RefCD.

### 4.5 EXTENSIONS TO WEAKLY SUPERVISED OBJECT DETECTION

To further verify the effectiveness of the FS loss, we extend RefCD to weakly supervised object detection. Specifically, we still do not use any category labels and only use box labels from COCO BASE for weakly supervised training of RefCD. Except for the different box labels utilized, other training settings are identical to the Implementation details in Section 4.1. As shown in Table 10 (1-st row), RefCD trained with weakly supervision achieves better performance on COCO NOVEL. Additionally, we explore experiments that use mask-level features for training. In other words, when calculating $p_n$ and $\hat{p}_n$, we employ mask-level annotations from COCO BASE to directly compute the pixel-wise feature similarity between two objects. As shown in Table 10 (2-nd row), RefCD trained with mask-level feature similarity outperforms the model trained with box-level feature similarity. (More details are in Section A.3)

### 4.6 EXTENSIONS TO UNSUPERVISED SINGLE OBJECT TRACKING

To further demonstrate the effectiveness of RefCD, we apply it to the single object tracking (SOT) task. Following the standard SOT experiment setting, we use the instance in the first frame as the reference image and continuously predict the corresponding instance in each subsequent frame, selecting the prediction with the highest confidence. As illustrated in Figure 5, we show the qualitative results on VOT 2018 Kristan et al. (2018) of RefCD and USOT Zheng et al. (2021). It can be observed that RefCD can handle various deformations of the instance. The confidence of the predicted instance varies with its shape changes but remains stable overall, which further demonstrates effectiveness and generalization ability of RefCD. Quantitative results are reported in Section A.8.

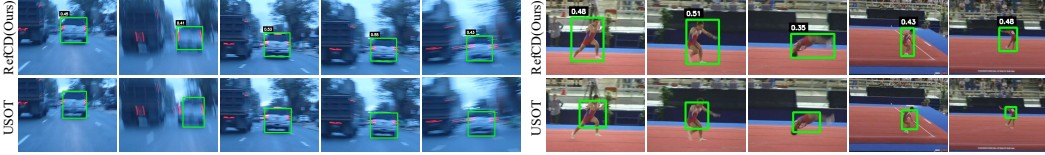

Figure 5: Qualitative unsuipervised single object tracking results of RefCD and USOT.

## 5 CONCLUSION

In this paper, we introduce a reference-driven strategy to train RefCD in a fully unsupervised manner. Instead of relying on explicit category labels, we introduce a feature similarity matching strategy to enable category discrimination in an unsupervised manner. To further enhance the learning of discriminative category features across different objects, we design a feature similarity loss. Furthermore, based on feature similarity matching, RefCD can also be applied to single object tracking task. For future work, we aim to improve the feature interaction mechanism to better model relationships between reference and detected objects, thereby further enhancing detection performance.

## ETHICS STATEMENT

Our work is entirely based on publicly available datasets in the field of object detection, all of which are openly accessible and ethically compliant. Specifically, our method does not involve human subjects, does not collect any personally identifiable information, and does not handle sensitive data. Therefore, issues such as privacy, informed consent, and data anonymization are not applicable to this work.

## REPRODUCIBILITY STATEMENT

We are committed to ensuring the reproducibility of our work and will make the complete source code, pre-trained models, and experimental scripts publicly available upon publication. To facilitate the review process, we provide detailed implementation details in both the main text and the appendix. All experimental details and model parameter settings are described in detail in the Section 4.1 of the main text. The complete model architecture and the full formulas of the loss functions have been elaborated in detail in the Method (Section 3) of the main text. In addition, the generation steps of pseudo boxes in the main text are provided in Appendix A.6.

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

# A APPENDIX

The appendix provides more details and results that are not included in the main paper due to space limitations. The contents are organized as follows:

- Section A.1 provides additional quantitative results of RefCD and presents some bad cases.
- Section A.2 presents the impact of training data volume on RefCD and analyzes the existing challenges in unsupervised object detection.
- Section A.3 introduces the extension of RefCD to weakly supervised learning, including the experimental setting of weakly supervised learning, the discussion on the quality of box labels for weakly supervised learning, and the exploration of difficult scenarios.
- Section A.4 presents 5-shot detection results and gives an experimental analysis.
- Section A.5 explains the difference between U2Seg and RefCD, and gives comparison on category-aware detection results on COCO val2017.
- Section A.6 presents the implement details of pseudo boxes generation and the specific working mechanism of the reference encoder.
- Section A.7 shows the reference images used in metric evaluation in this paper.
- Section A.8 provides quantitative results of RefCD on the single object tracking task.
- Section A.9 presents the experimental results and analysis of RefCD in domain-specific scenarios, demonstrating its generalization and stability.
- Section A.10 presents the qualitative results and analysis of RefCD on visually similar but semantically distinct objects.

## A.1 MORE VISUALIZATION RESULTS

In this section, we first present some failure cases and analyze their underlying causes. Then, we provide more visualizations to further demonstrate the effectiveness of RefCD.

**Failure case visualization and analysis.** We present some failure cases of RefCD in Figure 6. First, as shown in Figure 6(a), in crowded scenes, the semantic distinction between the foreground and background is ambiguous, and they share common visual features. Detectors fail to distinguish individual objects and detect all instances as a single object. Second, when multiple objects have a relative relationship (*e.g.*, a person riding a horse), the detector mistakenly treats them as a single entity, leading to incorrect category matching (Figure 6(b)). Finally, when objects are occluded, resulting in multiple disconnected image patch (Figure 6(c)), the detector fails to recognize them as a single object. We believe that low-quality category-agnostic detection is the primary factor affecting the detector's performance. Pseudo boxes generation fails to handle the scene in Figure 6, producing inaccurate pseudo boxes and leading to poor performance in these situations.

Furthermore, these low-quality category-agnostic detection results may lead to incorrect category-aware object matching. As shown in the 1-st row of Figure 7, erroneous detections exhibit similar feature similarities to the reference images of *person* and *horse*, making correct feature similarity matching impossible. Additionally, incomplete detections result in incorrect matching, as illustrated in the 2-nd row of Figure 7. Moreover, low-quality reference images can also degrade detection performance. If the detector struggles to extract useful information from the reference image, it will always produce incorrect detection results.

**More visualization and analysis.** As shown in Figure 11, we provide the visualization results of RefCD on the category-agnostic detection task. It can be observed that the proposed method can not only detect the objects annotated in COCO but also detect some novel category objects that are not annotated in the dataset, as shown by the red boxes in Figure 11. We also provide more visualizations of category-aware and category-agnostic detection results in Figure 16 and Figure 17. As shown, RefCD demonstrates effective performance in most scenarios.

**Visualization and Analysis of Reference Images Only Contain Partial Views of An Object.** Our detector faithfully detects objects similar to the reference images. As shown in Figure 8, detection results depend on reference images. If the reference image is a partial region of a car (*e.g.*, a wheel),

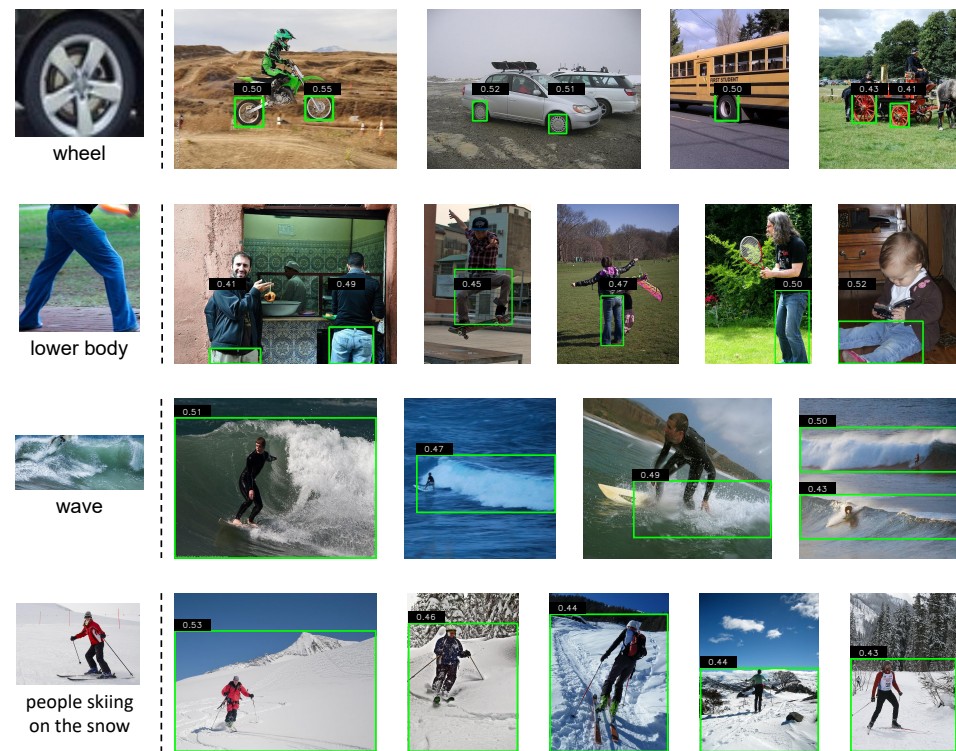

Figure 8: Visualization results of reference images only contain partial views of an object. The figure presents the detection results with *wheel*, *lower bodies*, *wave* and *people skiing on the snow* as reference images, and RefCD faithfully detects objects of interest in accordance with the reference images.

the detected objects will correspond to that target region (*i.e.*, the wheel part). Furthermore, the detection confidence reflects the similarity between the objects and the reference image. As illustrated in the Figure 8 1-st row, the similarity between carriage wheel and car wheel is lower than that between different car wheels.

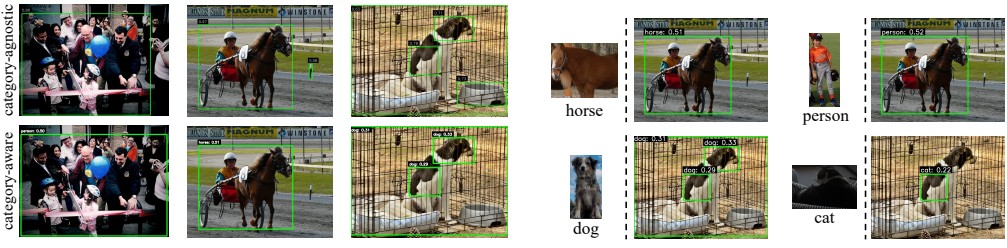

Figure 6: Visualization results of failure cases. The 1-st row show category-agnostic detection results, while the 2-nd row correspond to category-aware detection results.

Figure 7: Visualization results of failure cases. The impact of different reference images on category-aware object detection in the same scene.

## A.2 DISCUSSION ON TRAINING BOX VOLUME

As shown in Table 11, we analyze the impact of different amounts of training data (*i.e.*, the number of instances in the training set) on RefCD. Following previous work Wang et al. (2023a), we generated approximately 1,508,000 pseudo boxes from 1,300,000 images in ImageNet Deng et al. (2009) for training. The results in the lower part of Table 11 show the experimental performances of training RefCD with different numbers of pseudo boxes, corresponding to 30%, 60%, and 100%

of the data volume, respectively. The results indicate that as the number of pseudo boxes used for training increases, the performance of the detector also improves. With the same amount of data, the unsupervised trained RefCD outperforms most supervised detectors on the COCO NOVEL dataset. In addition, RefCD achieves performance comparable to SINE with only 1/8 of the training boxes, which demonstrates that category similarity learning can effectively leverage the advantages of unsupervised learning.

For a fair comparison under the same number of images and instances, we also provide the experimental results of RefCD trained on COCO BASE. As shown in the 4-th row of Table 11, we only use the bounding boxes of COCO BASE without category annotations for training, and the AP on COCO NOVEL reaches 24.9%, which is far higher than other supervised one-shot detection methods. This further demonstrates the superiority of RefCD.

We can generate only 1,500K pseudo boxes from 1,300K images, while the manually annotated Object365 dataset annotates 10,101K bounding boxes from just 638K images. Compared to unsupervised generation methods, manual annotation accurately annotates objects in dense and occluded scenes. This discrepancy implies that unsupervised training requires significantly more images than supervised methods. In future work, we will focus on enhance data utilization efficiency and improve the performance of unsupervised detectors.

Table 11: One-shot detection results with different training data volumes. Box Number represents the number of annotated bounding boxes in the dataset and refers to the number of generated pseudo boxes in RefCD.

| Method | Publication | Paradigm | Pretrain | | COCO NOVEL |
|---|---|---|---|---|---|
| | | | Dataset | Box Number | $AP_{box}$ |
| RefT Han et al. (2024) | TPAMI'24 | | COCO BASE | 500K | 5.2 |
| SINE Liu et al. (2024b) | NIPS'24 | Supervised | COCO BASE | 500K | 10.5 |
| SINE Liu et al. (2024b) | | | COCO+ADE20K+Object365 | 11704K | 18.0 |
| RefCD (Ours) | | | ImageNet | 452K | 8.9 |
| RefCD (Ours) | | Unsupervised | ImageNet | 905K | 12.3 |
| RefCD (Ours) | | | ImageNet | 1508K | **15.1** |

### A.3 EXTENSION ON WEAKLY SUPERVISED OBJECT DETECTION

Although RefCD achieves effective performance in unsupervised object detection, it is constrained by the quality of generated pseudo boxes. Specifically, it still struggles with occlusion and dense scenarios, as illustrated in Figures 6 and 7. Thus, we make some exploration in the weakly supervised domain. In this section, we first clarify the settings for weakly supervised training, then present experiments and analysis on the sensitivity of RefCD to box accuracy, and finally introduce two exploration methods for difficult cases.

#### A.3.1 DATASET AND SETTINGS

**COCO** is a standard object detection benchmark with 118K images across 80 categories. For openset one-shot evaluation, following Liu et al. (2024b), we split COCO into COCO NOVEL (20 categories overlapping with PASCAL VOC Everingham et al. (2010)) for evaluation, and COCO BASE (the remaining 60 categories serve as base categories) for training. Please note that we still do not use any category labels in weakly supervised training and only use the box labels of COCO BASE.

**Implement details.** The ViT-large model pre-trained with DINOv2 is used as the reference encoder and the detector is RT-DETR Zhao et al. (2024) with ResNet50 He et al. (2016) pre-trained with unsuerpvised method ReLiCV2 Tomasev et al. (2022) as the backbone. The model is trained on 2 NVIDIA RTX 3090 GPUs for 80K iterations with a batch size of 16, and an exponential moving average (EMA) with a decay rate of 0.9999 is applied. Hyperparameters $\alpha$ is set to 2, $\beta$ is set to 4, $K$ is set to 100, $\tau$ is set to 0, and temperature is set to 100.

### A.3.2 DISCUSSION ON TRAINING PSEUDO BOXES QUALITY

This section focuses on exploring the impact of unreliable bounding boxes on the performance of object detectors, with a specific emphasis on comparing the behavior of RefCD with other detection methods. To provide concrete evidence for this analysis, experimental results obtained from testing RefCD and other detectors under conditions of unreliable training data are presented in Tables 12 and 13.

As noted in the main paper, RefCD adopts a unique training strategy that leverages pseudo boxes generated on the ImageNet dataset. A key characteristic of this approach is that the process of generating these pseudo boxes, while valuable for training, is not without flaws and inevitably produces a certain number of unreliable results. As illustrated in Figure 9, in the context of traditional detection tasks, green boxes are used to denote correct annotations, whereas red boxes represent unreliable or incorrect ones. For RefCD, however, the criterion for a correct detection result is defined differently: a result is considered correct if it aligns with the reference pseudo box, regardless of whether that pseudo box would be deemed accurate in a traditional sense. This fundamental difference in evaluation criteria enables RefCD to effectively handle pseudo boxes that are generally regarded as low-quality in standard detection scenarios.

For experimental reproducibility and fairness, we use COCO BASE uniformly as training data. We introduce different influencing factors (Box Ratio and Noise Ratio) to the original annotations. Box Ratio refers to the proportion of boxes randomly retained from each image, simulating the incomplete recall issue in pseudo box generation. Noise Ratio represents the maximum proportional offset added to each box, simulating the inaccurate prediction issue in pseudo box generation. As shown in Table 12, supervised methods are more sensitive to box quality: as the quality of GT boxes degrades, their detection accuracy is hardly guaranteed. In contrast, RefCD makes better use of low-quality boxes through a more robust training strategy, yielding superior detection results. Table 13 shows that compared with CutLER, RefCD exhibits stronger robustness to unreliable boxes. With the same training data, it achieves better category-agnostic detection results.

Table 12: Results of category-aware detection trained with different interference factors added to the training data. **Box Ratio** represents the proportion of randomly selected boxes. **Noise Ratio** represents the maximum proportion of random offsets added to boxes. The detector is trained on COCO BASE and evaluated on COCO NOVEL.

| Box Ratio | Noise Ratio | Method | COCO NOVEL (category-aware) |
|---|---|---|---|
| | | | $AP_{50:95}$ |
| 1.0 | 0 | RefCD | 26.2 |
| | | SINE | 10.5 |
| | 0.2 | RefCD | 19.3 |
| | | SINE | 8.3 |
| | 0.4 | RefCD | 10.1 |
| | | SINE | 6.2 |
| 0.9 | 0 | RefCD | 24.3 |
| | | SINE | 10.1 |
| | 0.2 | RefCD | 18.2 |
| | | SINE | 7.5 |
| | 0.4 | RefCD | 7.8 |
| | | SINE | 4.3 |

Table 13: Results of category-agnostic detection trained with different interference factors added to the training data. **Box Ratio** represents the proportion of randomly selected boxes. **Noise Ratio** represents the maximum proportion of random offsets added to boxes. The detector is trained on COCO BASE and evaluated on COCO val2017.

| Box Ratio | Noise Ratio | Method | COCO val2017 (category-agnostic) | | |
|---|---|---|---|---|---|
| | | | $AP_{50:95}$ | $AP_{50}$ | AR |
| 1.0 | 0 | RefCD | 42.2 | 65.1 | 56.6 |
| | | CutLER | 37.0 | 60.7 | 52.3 |
| | 0.2 | RefCD | 28.2 | 47.7 | 44.1 |
| | | CutLER | 17.9 | 40.3 | 35.8 |
| | 0.4 | RefCD | 10.8 | 22.4 | 28.2 |
| | | CutLER | 7.9 | 19.0 | 26.8 |
| 0.9 | 0 | RefCD | 38.3 | 61.2 | 53.7 |
| | | CutLER | 35.2 | 59.7 | 49.5 |
| | 0.2 | RefCD | 25.7 | 43.4 | 40.9 |
| | | CutLER | 15.8 | 37.9 | 33.2 |
| | 0.4 | RefCD | 9.1 | 18.7 | 26.3 |
| | | CutLER | 6.5 | 17.4 | 24.9 |
| 0.7 | 0 | RefCD | 33.7 | 53.6 | 45.1 |
| | | CutLER | 28.6 | 49.9 | 41.3 |
| | 0.2 | RefCD | 22.6 | 34.8 | 37.1 |
| | | CutLER | 13.5 | 30.3 | 29.0 |
| | 0.4 | RefCD | 5.1 | 15.5 | 22.3 |
| | | CutLER | 2.6 | 9.5 | 19.3 |

### A.3.3 EXPLORATION OF DIFFICULT SCENARIOS

Figure 9: Different training strategy.

Table 14: Results of weakly supervised training RefCD with box label of COCO BASE, and evaluation on COCO NOVEL.

| Method | Paradigm | Feature Level | Output Layer | $AP_{box}$ |
|---|---|---|---|---|
| | | Mask | 6 | 28.3 |
| | | Box | 6 | 24.9 |
| RefCD | Weakly | Box | 5 | 24.5 |
| (Ours) | Supervised | Box | 4 | 24.0 |
| | | Box | 3 | 23.3 |
| | | Box | 2 | 22.2 |
| | | Box | 1 | 19.0 |

**Mask-level similarity.** In some occlusion and crowded scenarios, box-level features often fail to accurately represent object features, as bounding boxes may include excessive background or other objects. To enable the detector to focus on the own feature of the instance, we use mask-level features for the weakly supervised training of RefCD. Specifically, during training, we utilize mask-level (*i.e.*, pixel-wise) features to calculate the similarity $p_n$ between the reference image and other objects in the image. Similarly, we use the mask-level features of the reference image to compute the similarity $\hat{p}_n$ with the predicted query. We directly adopt the built-in mask annotations from the COCO dataset. During input, we perform a dot product operation between the mask and box of the instance, and then feed the result into the reference encoder to obtain the mask-level features of the instance. As presented in Table 14, RefCD trained with mask-level features for weak supervision achieves better category-aware detection results on COCO NOVEL. This demonstrates that fine-grained features are more conducive to the learning of feature similarity, while also verifying the effectiveness of the FS loss. We present the visualization results of training with mask-level feature similarity in Figure 10. As illustrated, compared with unsupervised-trained RefCD, weakly supervised training with higher-quality boxes enables more effective handling of crowded and occlusion scenarios. Additionally, mask-level feature similarity assists the detector in better understanding the objects of interest.

**Box-refinement mechanisms.** RefCD uses multi-layer decoders to improve localization. We use 6 Transformer layers as decoder in RefCD. We conduct experiments on RefCD (weakly supervised trained on COCO BASE) and provide results for each decoder layer. As shown in Table 14 rows 2–7, iterative box-refinement mechanisms help improve detection performance. The 6-th decoder layer outputs the best detection results.

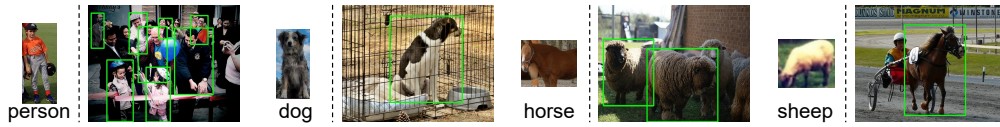

Figure 10: Visualization results of weakly supervised training RefCD on COCO NOVEL.

## A.4 RESULTS OF FIVE-SHOT DETECTION

We first conduct a comprehensive comparison between the proposed method and the existing five-shot detection methods, aiming to systematically evaluate the performance advantages of our ap-

proach in the five-shot detection scenario. This comparison is crucial as it helps to position our method within the current state-of-the-art and highlight its unique contributions.

The evaluated methods include several representative ones in the field: Meta R-CNN Yan et al. (2019), MTFA Ganea et al. (2021), iFS R-CNN Nguyen & Todorovic (2022), Meta-DETR Zhang et al. (2022a), RefT Han et al. (2024), GlobalTrack Huang et al. (2020), and SINE Liu et al. (2024b). It is important to note that all these existing five-shot detection methods rely on supervised training, which means they require labeled data with accurate annotations for the training process. In contrast, our proposed method may have distinct training characteristics that set it apart, and this difference is worth considering in the comparison.

As shown in Table 15, we compare the proposed method with various five-shot detection methods. It can be observed that RefCD also achieves the best performance in Five-shot detection. This superior performance is not only reflected in the evaluation metrics in the table. Compared with existing methods, RefCD also has advantages in addressing the challenges of five-shot detection, such as requiring limited training samples and being able to quickly adapt to new categories.

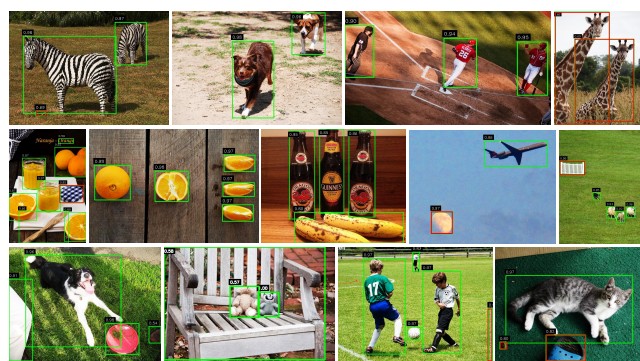

Table 15: Five-shot detection results (category-aware) on COCO NOVEL. The results of existing methods on COCO NOVEL are accessed from their papers for the reason that some methods are not publicly available.

| Methods | $AP$ |
|---|---|
| *pre-defined categories methods* | |
| Meta R-CNN Yan et al. (2019) | 3.5 |
| MTFA Ganea et al. (2021) | 6.6 |
| iMTFA Ganea et al. (2021) | 6.2 |
| iFS R-CNN Nguyen & Todorovic (2022) | 10.5 |
| Meta-DETR Zhang et al. (2022a) | 15.4 |
| *reference-based methods* | |
| SINE Han et al. (2024) | 16.0 |
| RefCD(Ours) | 17.7 |

Figure 11: Qualitative category-agnostic results of RefCD. Red boxes indicates objects detected by RefCD that are either unannotated or incorrectly annotated in COCO.

## A.5 COMPARISON WITH U2SEG

In this section, we present a detailed comparison between RefCD and U2Seg Niu et al. (2024). We first introduce the working mechanism of U2Seg, followed by a detailed analysis of the differences between the two methods. Finally, we provide their category-aware detection results on COCO val2017.

### A.5.1 INTRODUCTION OF U2SEG

Following previous unsupervised method Wang et al. (2023a), U2Seg generates pseudo boxes/masks on ImageNet Deng et al. (2009) without category labels. Unlike other methods, U2Seg extracts features from these pseudo boxes using DINOv2 Oquab et al. (2023) and performs feature clustering. It manually sets the number of cluster centers $N$ as pseudo labels for the pseudo boxes and trains the detector using both pseudo boxes and pseudo labels. During inference, U2Seg applies Hungarian matching Carion et al. (2020) between pseudo labels and category names in the real dataset, mapping the test results to manually annotated category labels.

Table 16: Results of different category-aware unsupervised methods. CutLER+ refers to training with feature clustering methods.

| Method | COCO val2017 | | | |
|---|---|---|---|---|
| | $AP$ | $AP_{50}$ | $AP_{75}$ | $AR$ |
| *zero-shot methods* | | | | |
| CutLER+ Wang et al. (2023a) | 5.9 | 9.0 | 6.1 | 10.3 |
| U2Seg Niu et al. (2024) | 7.3 | 11.8 | 7.5 | 21.5 |
| *one-shot methods* | | | | |
| RefCD (Ours) | 10.6 | 18.7 | 9.9 | 24.3 |

### A.5.2 DIFFERENCE BETWEEN U2SEG AND REFCD

As the number of clusters $N$ increases, the detector trained with U2Seg can recognize more categories. However, the pseudo label clustering method is inherently constrained by the object categories in the training set, making it incapable of detecting novel categories absent from the training data. To address this limitation, we propose RefCD, which discards explicit category label constraints and introduces a category-aware detection method based on feature similarity matching. As described in the main paper, RefCD performs open-set object detection by leveraging reference images and feature similarity matching. Compared to U2Seg, RefCD better mitigates domain gaps between datasets and achieves effective novel category detection.

### A.5.3 CATEGORY-AWARE DETECTION RESULTS

We present the 80 categories detection results of U2Seg, CutLER, and RefCD on the COCO Lin (2014) val2017 dataset in Table 16. Since U2Seg does not provide runnable evaluation code, we report the results given in its original paper Niu et al. (2024).

**Please note** that the goal of this experiment does demonstrate better category-aware object detection performance of RefCD over U2Seg. A strictly fair comparison is not feasible. Existing unsupervised methods (*e.g.*, U2Seg and CutLER) are not designed for one-shot detection, whereas RefCD performs category-aware one-shot detection using reference images. We hope to foster the development of category-aware unsupervised object detection rather than to establish a direct superiority of one approach over another.

## A.6 DETAILS OF PSEUDO-BOX GENERATION AND REFERENCE ENCODER

### A.6.1 PSEUDO-BOX GENERATION BASED ON MASKCUT

**Preliminaries.** MaskCut Wang et al. (2023a) extends the TokenCut Wang et al. (2023b), which is limited to discovering a single object per image, to enable multiple object discovery. It starts by constructing a patch-wise similarity matrix using attention map from self-supervised DINO Caron et al. (2021). Then, NCut is applied to this matrix to obtain a bipartition $x^t$, from which a binary mask $M^t$ is created

$$M_{ij}^t = \begin{cases} 1, & M_{ij}^t \geq \text{mean}(x^t) \\ 0, & \text{else.} \end{cases} \tag{9}$$

To determine the foreground, two criteria are used: the foreground mask should contain the patch corresponding to the maximum absolute value in the second smallest eigenvector $M^t$, and the foreground should have less than two of the four corners. If criteria are not met, the partition of foreground and background is reversed ($M_{ij}^t = 1 - M_{ij}^t$). For subsequent objects, the node similarity $W_{ij}^{t+1}$ is updated by masking out nodes corresponding to the foreground from previous stages, using

$$W_{ij}^{t+1} = \frac{(K_i \prod_{s=1}^t \hat{M}_{ij}^t)(K_j \prod_{s=1}^t \hat{M}_{ij}^t)}{\|K_i\|_2 \|K_j\|_2} (\hat{M}_{ij}^t = 1 - M_{ij}^t), \tag{10}$$

where $K_i, K_j$ is feature vectors of image patches $i$ and $j$ extracted by DINO and $\hat{M}_{ij}^t$ is a mask indicator where $\hat{M}_{ij}^t = 1 - M_{ij}^t$, used to mask out nodes corresponding to the foreground from previous stages. When generating pseudo boxes, MaskCut repeats the process of NCut and mask creation $N$ times.

To generate pseudo boxes for object instances, we utilize the binary masks $M^t$ produced by MaskCut. For each mask $M^t$, we first find the minimum $x_{\min}$ and maximum $x_{\max}$ $x$ - coordinates, and the minimum $y_{\min}$ and maximum $y_{\max}$ $y$ - coordinates of the foreground pixels (where $M_{ij}^t = 1$).

The pseudo box $B^t$ is defined as:

$$B^t = (x_{\min}, y_{\min}, x_{\max} - x_{\min}, y_{\max} - y_{\min}), \tag{11}$$

where $(x_{\min}, y_{\min})$ as the top-left corner, and $x_{\max} - x_{\min}$ and $y_{\max} - y_{\min}$ as the width and height.

**Implement Details.** We follow the publicly accessible codebase of CutLER Wang et al. (2023a) to generate pseudo boxes on the ImageNet dataset Deng et al. (2009). These pseudo boxes serve as the foundation for conducting unsupervised training on the RT-DETR Zhao et al. (2024). Specifically, we carry out one round of self-supervised training with RT-DETR, leveraging the pseudo boxes.

Subsequently, the pre-trained RT-DETR model is employed to perform inference on ImageNet. During this inference process, bounding boxes whose confidence scores exceed a threshold $\gamma$ are selectively retained. These retained bounding boxes then constitute the final training data, which can be utilized for further model refinement or other relevant training tasks. During the experiment, we set $N$ to 3 and $\gamma$ to 0.9.

**Visualization of pseudo-boxes.** Figure 12 presents the visualization results of the generated pseudo-boxes. As can be observed, most of the generated pseudo-boxes fail to accurately localize objects with clear categorical meanings, and even for some objects with clear categorical meanings, accurate bounding boxes cannot be obtained.

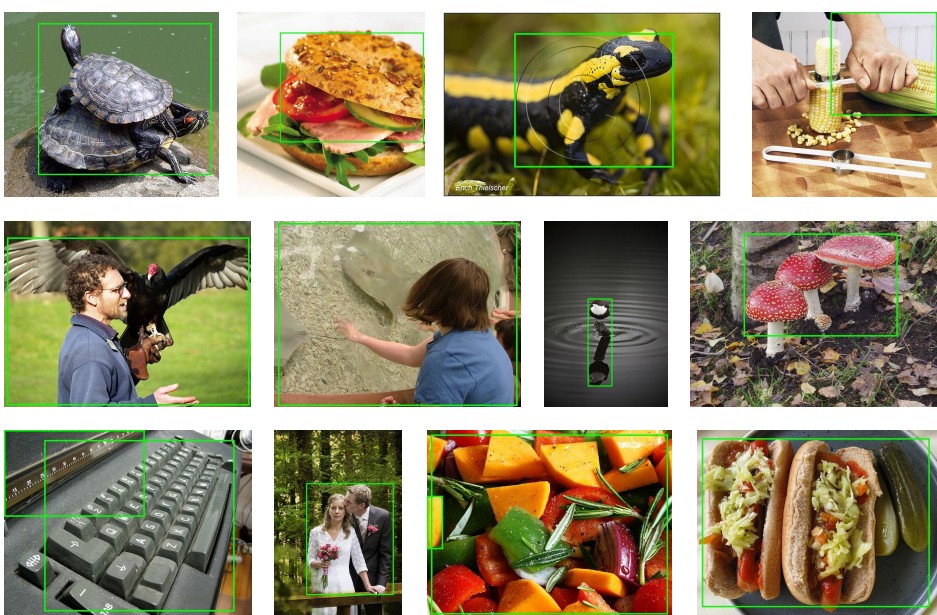

Figure 12: Visualization of generated pseudo-boxes on ImageNet.

### A.6.2 DETAILS OF THE REFERENCE ENCODER.

**Implementation method.** RefCD adopts the self-supervised pre-trained DINOv2 Oquab et al. (2023) as the reference encoder, which parameters are frozen. The reference encoder performs category feature embedding on objects during both training and inference. During training, we randomly select pseudo boxes as reference images and calculate training loss based on the similarity of feature embeddings. To reduce the cost of repeated calculations, we use DINOv2 to conduct offline preprocessing on all pseudo boxes in the training set. Specifically, after pseudo boxes generation, we input these pseudo boxes into DINOv2 to obtain their category embeddings and save them locally. The corresponding embeddings are directly read based on image IDs for calculation in each iteration. It should be noted that during inference, the reference encoder performs real-time embeddings of reference images without data preprocessing.

**Implementation details.** The ViT-large model pre-trained with DINOv2 is used as the reference encoder. We adopt the cls token vector output by DINOv2 as the category feature embedding, with the feature dimension set to 1024. In the data preprocessing stage, we generate local category feature embeddings for all pseudo boxes. Experiments are conducted on 2 RTX 3090 GPUs. The input image size is set to $560 \times 560$ (consistent with the inference stage), the batch size is set to 64, and the total running time is 20 hours. Please note that DINOv2 is a self-supervised trained feature extractor. The pre-trained model used in RefCD has not been trained on any category-labeled data. Furthermore, the classification model of DINOv2 requires additional post-training on ImageNet with a classification head. RefCD does not perform this step, which fully complies with the definition and requirements of unsupervised training.

**Generalizability on Custom datasets.** The proposed RefCD is generally applicable to most custom datasets. Its adaptability stems from the strong representational capabilities of models DINOv2. DINOv2 has been pre-trained on a large-scale dataset containing 142 million images. It can learn general visual features that are not limited to specific image domains. For most common object detection scenarios, even if the backbone network has not been pre-trained on relevant images, the reference encoder in RefCD can still capture high-quality category features. This means that for common datasets such as COCO Lin (2014), ImageNet Deng et al. (2009), and Pascal VOC Everingham et al. (2010), RefCD can be directly applied without training the reference encoder. However, for highly specialized custom datasets (e.g., medical images of rare diseases with unique anatomical features, or industrial defect detection data with unconventional textures), the fixed pre-trained reference encoder may fail to match domain-specific features. In such cases, the reference encoder can be unfrozen and fine-tuned on the custom dataset. In essence, the applicability of RefCD to custom datasets depends on scenario-specific adjustments: it can be used "out-of-the-box" in general scenarios, while in special scenarios, targeted optimization of the reference encoder is required.

## A.7 REFERNECE IMAGES

Figure 15 shows the 4 default groups of reference images. It can be observed that there are differences within the same category, thus necessitating a discussion on the impact of different reference images. For this purpose, we present the detection results of different reference images in Table 5 of the main text. It can be seen that although different sets of reference images yield varying detection performances, these differences are negligible, which demonstrates the generalization ability of RefCD on reference images. A carefully selected set of template images may yield better performance, but this is not our original intention. We agree that high-quality reference images can improve detection accuracy. However, in reference-based detection, how to stably select high-quality reference images remains a challenge for future research.

## A.8 EXTENSION REFCD ON SINGLE OBJECT TRACKING TASK

We report the quantitative results of RefCD on the unsupervised single object tracking (SOT) task, as presented in Table 17. We conducted the SOT task evaluation on the VOT 2018 Kristan et al. (2018) dataset. It can be observed that RefCD achieves comparable performance on the SOT task compared to KCF Henriques et al. (2014) and USOT Zheng et al. (2021). KCF is a traditional tracker that achieves real-time object tracking based on kernelized correlation filtering. USOT leverages unsupervised optical flow and dynamic programming to sample continuously moving objects for tracker training.

Table 17: Results of single object tracking.

| Method | $A \uparrow$ | $R \downarrow$ | $EAO \uparrow$ |
| --- | --- | --- | --- |
| KCF Henriques et al. (2014) | 0.447 | 0.773 | 0.135 |
| USOT Zheng et al. (2021) | 0.564 | 0.435 | 0.290 |
| RefCD(Ours) | 0.511 | 0.577 | 0.187 |

## A.9 GENERALIZATION IN DOMAIN-SPECIFIC SCENARIOS

RefCD enables category-aware detection based on feature similarity between objects without the need to use category features to predict pre-defined category labels. Thus, Domain-specific scenarios do not hinder its generalization. Table 18 and Figure 13 present the experimental results for industrial and underwater scenarios. The industrial scenario is selected from the RAD dataset Cheng et al. (2024), comprising 4 categories and 1224 industrial scene images. The underwater scenario is selected from the 3D-ZeF20 dataset Pedersen et al. (2020), with 'zebra fish' as the objects of interest. Since the test set does not provide ground-truth, we random chose 900 images from the training set for one-shot evaluation. As shown in Table 18, RefCD still enables effective detection in domain-specific scenarios. Although the scenes of RAD and MOT Challenge are highly special-

ized, the object complexity within these scenes is lower than that of COCO NOVEL, so that RefCD achieves even better performance in these scenarios. Notably, RefCD is pre-trained on ImageNet with a frozen reference encoder. This aims to demonstrate RefCD's generalization and stability.

Table 18: Results in domain-specific scenarios.

| Scenarios | Dataset | $AP_{50}$ | $AP$ | $AR$ |
|---|---|---|---|---|
| industry | RAD Cheng et al. (2024) | 44.3 | 30.7 | 57.2 |
| underwater | 3D-ZeF20 Pedersen et al. (2020) | 50.0 | 28.1 | 61.0 |

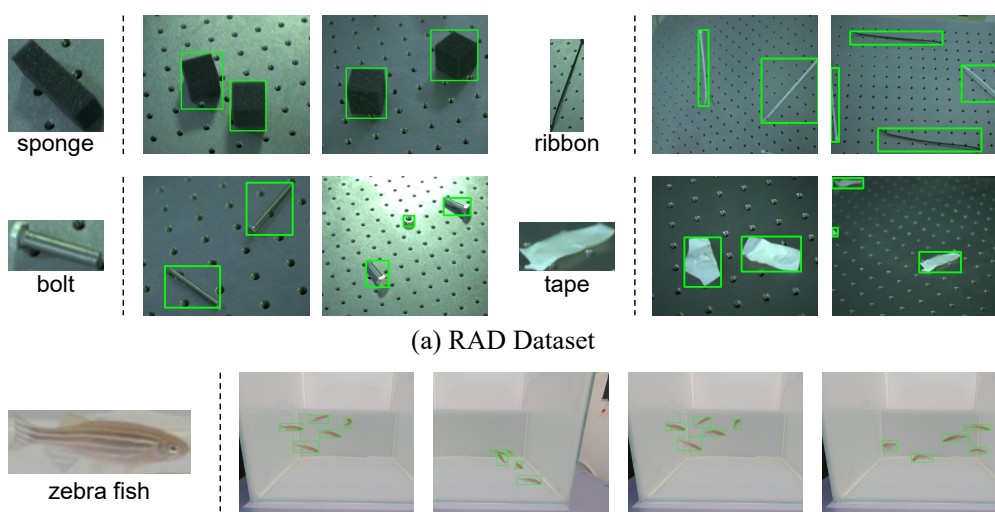

(a) RAD Dataset

(b) 3D-ZeF20 Dataset

Figure 13: Visualization results of domain-specific scenarios.

### A.10 RESULTS OF VISUALLY SIMILAR BUT SEMANTICALLY DIFFERENT OBJECTS

Based on manually set similarity thresholds, RefCD shows robustness to intra-class variants, as illustrated in Figure 4. However, in addition to intra-class variations, visually similar but semantically distinct objects present another challenge for existing detectors. A similar limitation exists for both supervised and unsupervised detectors, as shown in Figure 14(a), DETR, which is supervised-trained on the COCO dataset, also struggles to distinguish between orange apples and oranges. To further verify the generalization ability of RefCD, we report the qualitative experimental results of Siamese-DETR Liu et al. (2024a), DETR Carion et al. (2020), and RefCD on relevant cases (*i.e.*, orange apples and oranges). As shown in Figure 14(a) and Figure 14(b), RefCD can distinguish between visually similar but semantically distinct objects without causing noticeable category drift, while supervised-trained DETR cannot distinguish between orange apples and oranges.

However, its ability to handle such cases is limited. As illustrated in Figure 14(c) and Figure 14(d), although RefCD and Siamese-DETR can distinguish between orange apples and oranges to a certain extent, both orange apples and oranges exhibit almost the same similarity (with a difference of about 0.1) to the reference image (*i.e.*, either orange apple or orange). Matched objects typically yield higher feature similarity, enabling distinction between them using more refined similarity thresholds. Note that the difference in the confidence scores output by Siamese-DETR and RefCD stems from different confidence calculation methods rather than differences in model performance.

Case 1 is a real image selected from the COCO dataset Lin (2014). To conduct experiments in a scenario more consistent with the description, the image shown in Case 2 was generated using Doubao with the prompt: *"Generate an apple and an orange together, but both the orange and the apple are orange in color"*.

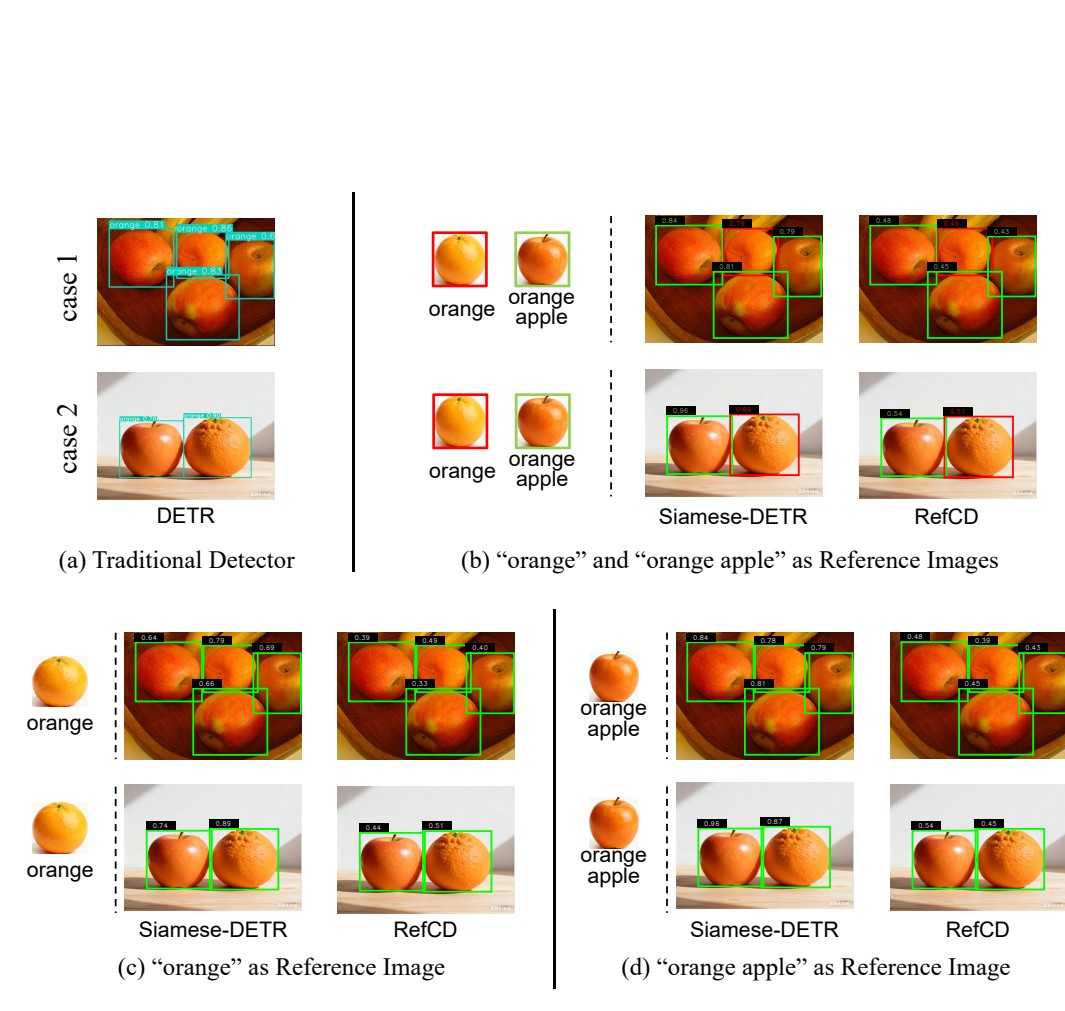

Figure 14: Visualization of visually similar but semantically distinct objects. (a) Detection results of DETR Carion et al. (2020) without reference images. (b) Detection results of Siamese-DETR Liu et al. (2024a) and RefCD using oranges as reference images. (c) Detection results of Siamese-DETR and RefCD using orange apples as reference images. **Please note that Case 1 is an image selected from the COCO dataset. To obtain images more consistent with the description, the image shown in Case 2 is generated by Doubao.**

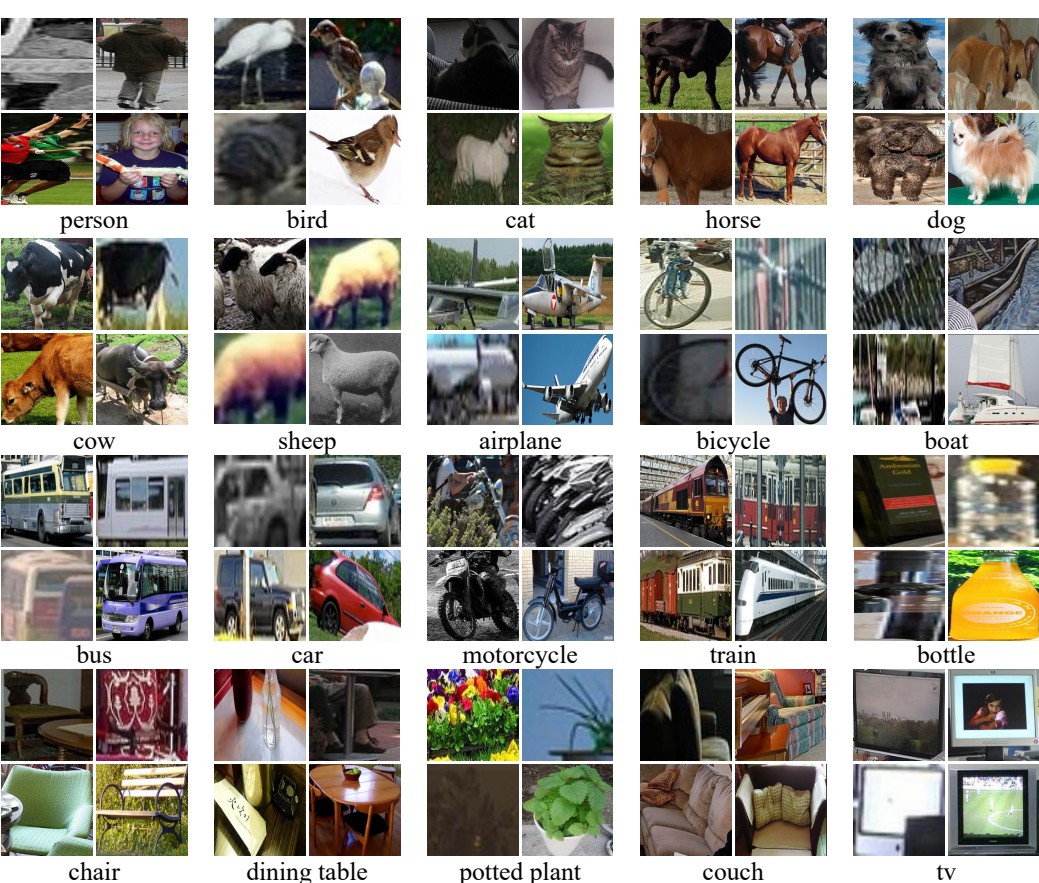

Figure 15: Template images used for each category. We present 4 template images used for each category, three of which are from COCO train2017, and the reference image at the bottom-right corner is from ImageNet. All template images are padded to square resolution.

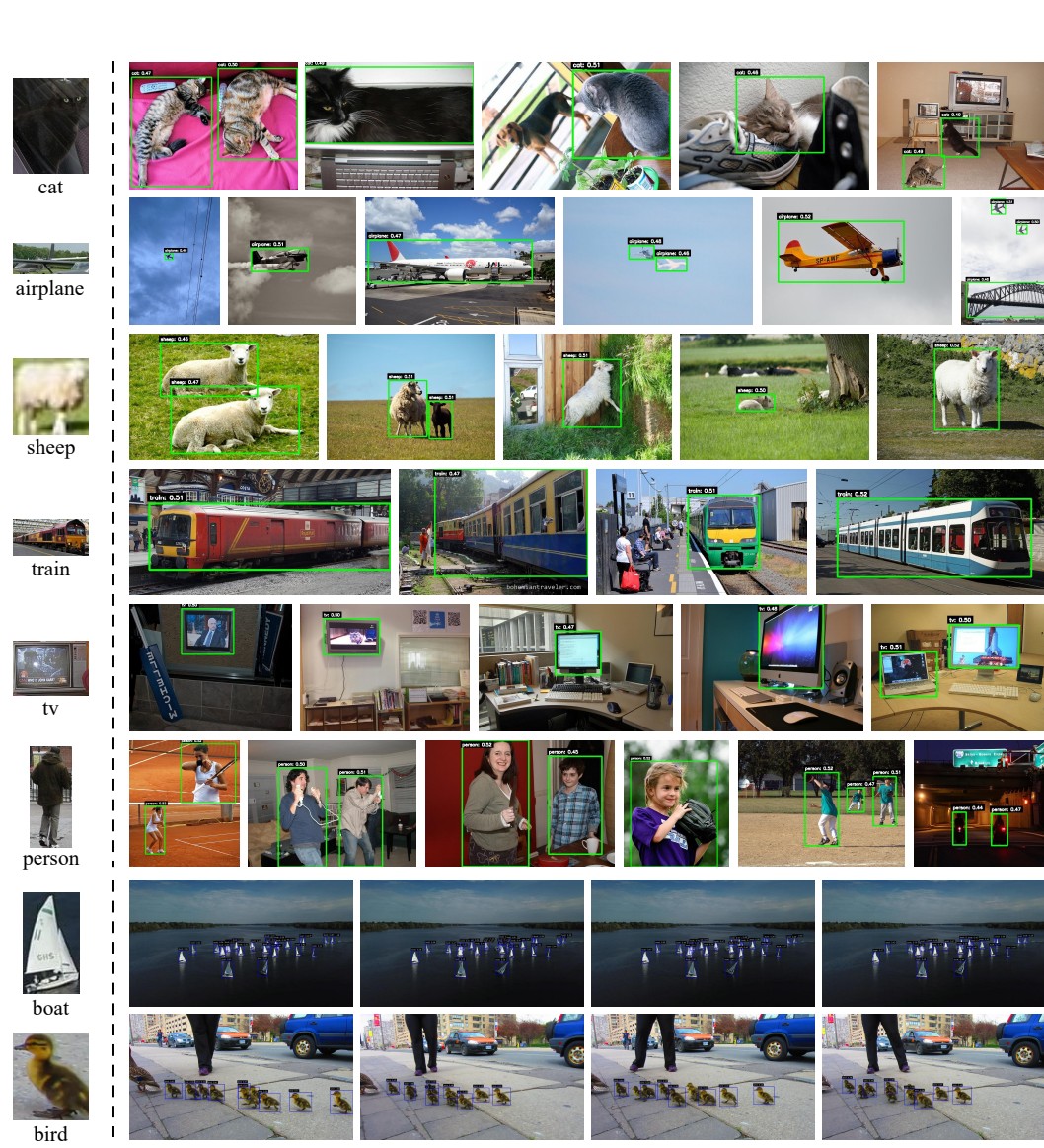

Figure 16: Visualization of category-aware detection on COCO NOVEL and GMOT-40. The reference images used for detection are shown on the left side of the dashed line. The top 6 sets of images present detection results on COCO NOVEL, while the bottom 2 sets display results on GMOT-40. We visualize all predictions with a confidence score greater than 0.3. Please note that to facilitate understanding, we add category names to the detection boxes. In actual inference, it is unnecessary to predefine categories for reference images. (Best viewed with zoom)

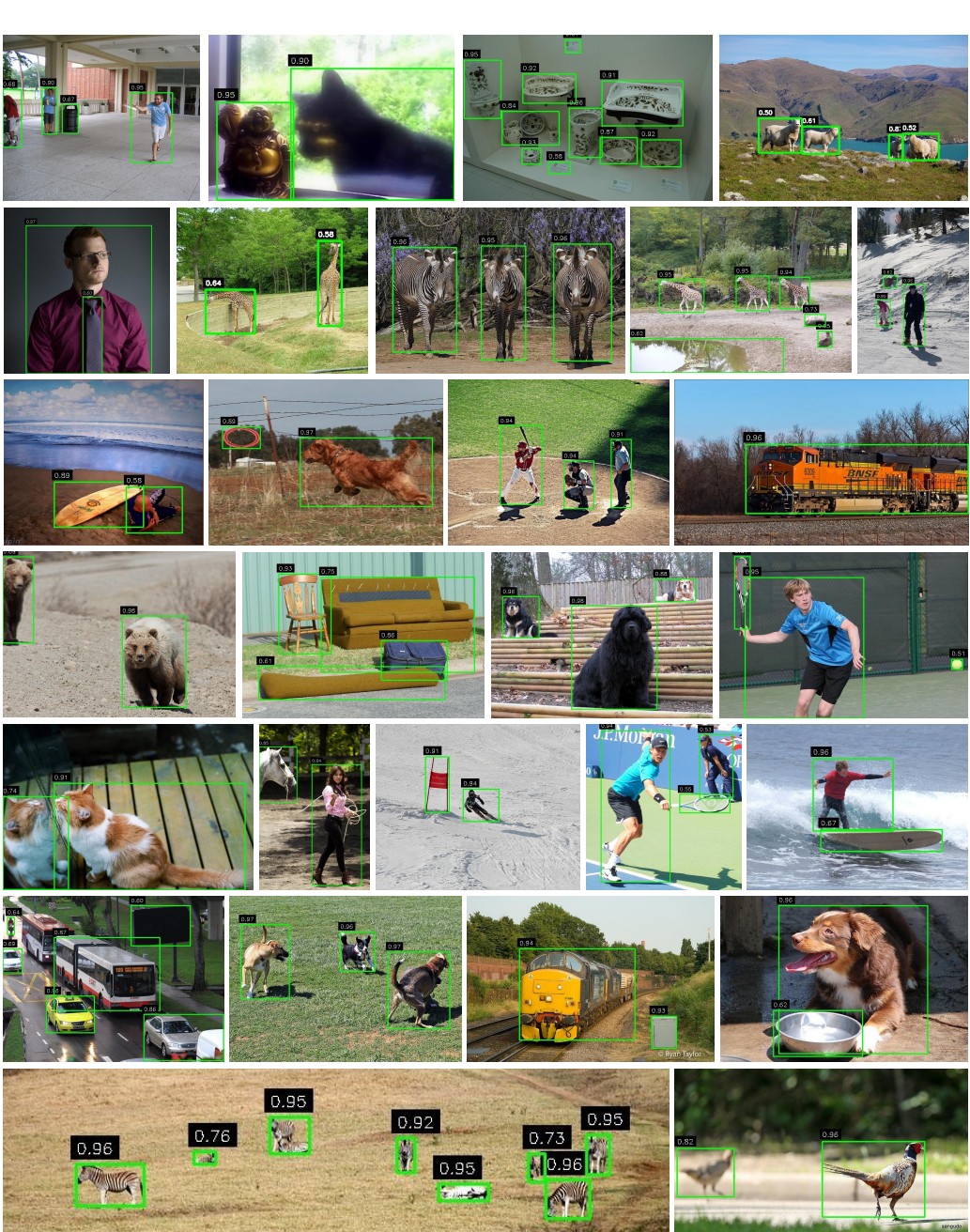

Figure 17: Visualization of category-agnostic detection on COCO val2017. Visualization results with a confidence score greater than 0.5 are provided. (Best viewed with zoom)

## STATEMENT ON LARGE LANGUAGE MODEL (LLM) USAGE

In the process of writing this paper, we utilized Large Language Models (LLMs) to support specific technical tasks, with their roles limited to Chinese-English translation of draft content and polishing of partial text (e.g., refining language fluency and standardizing academic expression). The LLMs involved in providing this assistance are Doubao, Qwen, and ChatGPT. Their usage was confined to auxiliary language processing, without contributing to research ideation, core argument development, data analysis, or the generation of key academic content. We confirm that we take full responsibility for the entire content of this paper, including the accuracy, authenticity, and academic integrity of the parts supported by LLM assistance. No content generated or polished by the aforementioned LLMs constitutes plagiarism, fabrication of facts, or any other form of scientific misconduct.

