# OpenReview forum: "Reference-based Category Discovery: Unsupervised Object Detection with Category Awareness"
_ICLR.cc/2026/Conference — Submitted to ICLR 2026_

### Official Review · Reviewer_4kU3 · 2025-10-26

**Soundness:** 2
**Presentation:** 2
**Contribution:** 2
**Rating:** 4
**Confidence:** 4

**Summary:**

The submission focused on the task of unsupervised object detection, and proposed Reference-based Category Discovery. Specifically, the proposed RefCD achieves category-aware detection without labels. Besides, the authors propose a feature similarity loss to encourage the detector to mine the categor information. The experiments are conducted on two large-scale benchmark datasets, which indicate the effectiveness of the proposed method.

**Strengths:**

1. The task of unsupervised object detection is interesting and fundmental in the field of computer vision.

2. The idea of building amodal segmentor based on SAM is reasonable and make sense.

3. The performances of proposed method are shown on various benchmark datasets, and outperform baselines by a large margin.

**Weaknesses:**

1. The description is confuzing in Line 40, what's the relationship between one-shot detection and unsupervised object detection? Do the authors focus on one-shot detection? The task should be consistently and clearly defined.

2. The task motivation and task-wise comparison are unclear. Compared with One-shot Detector, RefCD also uses reference images, but why is it without class label? What's the difference between the reference images in two tasks? More distinctions should be presented, e.g., including detailed training settings and test settings.

3. Why not train the reference encoder?

4. Is there any more recent baseline in single object tracking?

5. Closely related works focused on learning novel classes with category-agnostic proposal and similarity loss could be complemented to enrich the related works. The similar frameworks are effective supports for method design.

   > 1. "Weak-shot semantic segmentation via dual similarity transfer." Advances in Neural Information Processing Systems 35 (2022): 32525-32536.
   >
   > 2. "Weak-shot fine-grained classification via similarity transfer." Advances in Neural Information Processing Systems 34 (2021): 7306-7318.

6. Some typos found in Line 76 and Line 252.

**Questions:**

See Weakness.

---

> ### Author Response · Authors · 2025-11-20
> **Authors' Rebuttal 1**
>
> **Dear Reviewer 4kU3:**
>
> **We greatly appreciate your detailed feedback and the time you spent reviewing our work. We sincerely hope our response will address your questions about our work.**
>
> **W1: Definition of our work**
>
> >The description is confuzing in Line 40, what's the relationship between one-shot detection and unsupervised object detection? Do the authors focus on one-shot detection? The task should be consistently and clearly defined.
>
> We deeply apologize for the confusion caused by the inappropriate phrasing "focus on one-shot detection" in Line 40, it is an unintentional error. We have carefully revised the manuscript and clarified the relationship between our work and one-shot detection, as well as our core focus.
>
> Our RefCD primarily focuses on unsupervised object detection, one-shot detection methods are only introduced for quantitative comparison.
> Specifically, existing unsupervised methods fail to generate category labels or learn category-aware features, while one-shot detectors rely on accurate manual category labels for sample assignment and classification loss calculation, making them inapplicable to unsupervised scenarios without labels.
>
> To address the key limitation of "lack of category-aware learning in unsupervised detection," we propose RefCD: a reference image-based unsupervised framework that uses unlabeled reference images as semantic prompts and designs a Feature Similarity (FS) loss to guide category-aware feature learning from unlabeled pseudo-boxes.
>
> RefCD differs fundamentally from one-shot detection in training process and settings.
> RefCD does not require manually defined category labels.
> Moreover, the learning purposes for category-aware features are completely different (feature similarity vs. category labels).
> Due to the lack of unsupervised baselines for category-aware detection, we adopt supervised one-shot detection’s experimental settings for evaluation.
> As shown in Table 2, unsupervised RefCD outperforms most supervised one-shot methods.
>
>
> **W2: More detailed differences between two tasks.**
>
> >The task motivation and task-wise comparison are unclear. Compared with One-shot Detector, RefCD also uses reference images, but why is it without class label? What's the difference between the reference images in two tasks? More distinctions should be presented, e.g., including detailed training settings and test settings.
>
> We apologize for the confusion.
>
> **(1) Regarding category labels.**
>
> Category labels refer to manually predefined annotations for different object categories.
> One-shot detectors require such labeled data during training for sample assignment and loss calculation.
> In contrast, RefCD is an unsupervised framework designed to eliminate reliance on large-scale fine-grained annotations and reduce manual cost/time.
> It achieves category-aware detection by feature similarity between objects, requiring no category labels throughout training.
> Therefore, the "without class label" means RefCD does not use any category labels during training.
>
> For unified quantitative evaluation, we define the grounded categories of reference images during the evaluation process.This enables comparison with other methods.
>
> **(2) Difference in reference images.**
>
> In the training phase, due to different network structures and training paradigms, reference images play completely different roles.
> Reference images in one-shot detectors are used for one-shot fine-tuning or image feature enhancement, with final category labels predicted by a classifier (dependent on pre-defined labels).
> Reference images in RefCD are directly utilized for feature similarity calculation, guiding identification of objects of interest without label constraints.
>
> In the inference phase, the reference images that two tasks input are the same.
>
> **(3) Training and evaluation settings.**
>
> Training: RefCD adopts unsupervised training (without manually annotated category/box labels except generated pseudo-boxes).
>
> Evaluation: We randomly select 4 reference image sets (from COCO and ImageNet) and evaluate on the COCO NOVEL dataset, reporting average results.
> For unified quantitative comparison with other methods, we use identical reference images for the comparison, and define fixed categories for the reference images during evaluation (only for benchmarking, not training).

---

> ### Author Response · Authors · 2025-11-20
> **Authors' Rebuttal 2**
>
> **W3: Training of reference encoder**
>
> >Why not train the reference encoder?
>
> We choose not to train the reference encoder for two key reasons:
>
> First, DINOv2 (our reference encoder) is a highly generalized feature extractor pre-trained on large-scale data, delivering robust performance across most scenarios.
>
> Second, RefCD only relies on feature embeddings for similarity calculation, which requires no task-specific semantic information. Our goal is to enable open-set object detection across multiple scenarios. Freezing the reference encoder during training prevents overfitting to the training data, ensuring its generalization ability is retained for cross-scenario adaptation.
>
> **W4: Comparison of SOT task**
>
> >Is there any more recent baseline in single object tracking?
>
> Thank you for the insightful comments.
>
> Based on our search results, USOT [3] is the state-of-the-art method for unsupervised single object tracking. Most recent methods are supervised-trained trackers, so we select USOT [3] for comparison.
> In addition, we also compare with KCF [2], which is a traditional method that does not require deep learning.
> We report the quantitative results of RefCD on the unsupervised single object tracking (SOT) task, as presented in R-Table 6 (Table 9 of main paper).
> The SOT task evaluation is conducted on the VOT 2018 [1] dataset.
> It can be observed that RefCD achieves comparable performance on the SOT task compared to KCF [2] and USOT [3].
>
> [1] Kristan, Matej, et al. "The sixth visual object tracking vot2018 challenge results." Proceedings of the European Conference on Computer Vision (ECCV) workshops. 2018.
> [2] Henriques, João F., et al. "High-speed tracking with kernelized correlation filters." IEEE Transactions on Pattern Analysis and Machine Intelligence 37.3 (2014): 583-596.
> [3] Zheng, Jilai, et al. "Learning to track objects from unlabeled videos." Proceedings of the IEEE International Conference on Computer Vision. 2021.
>
> ### R-Table 6: Results of single object tracking.
> | Method        | A ↑   | R ↓   | EAO ↑ |
> | :-----        | :---: | :---: | :---: |
> | KCF [2]       | 0.447 | 0.773 | 0.135 |
> | USOT [3]      | 0.564 | 0.435 | 0.290 |
> | RefCD(Ours)   | 0.511 | 0.577 | 0.187 |
>
> **W5: Supplement to related works**
>
> >Closely related works focused on learning novel classes with category-agnostic proposal and similarity loss could be complemented to enrich the related works. The similar frameworks are effective supports for method design.
>
> Thank you for sharing these related works.
>
> We have supplemented related work sections with discussions of closely related studies.
> These related works propose to learning novel categories with similarity transfer [1] and complementary loss [2] provide effective support for method design. These works provide valuable theoretical support for our method design, enriching theacademic foundation of our paper.
>
>
> [1] "Weak-shot semantic segmentation via dual similarity transfer." Advances in Neural Information Processing Systems 35 (2022): 32525-32536.
> [2] "Weak-shot fine-grained classification via similarity transfer." Advances in Neural Information Processing Systems 34 (2021): 7306-7318.
>
> **W6: Writing problems**
>
> >Some typos found in Line 76 and Line 252.
>
> Thank you for checking on that level of detail!
> We have carefully revised the typos, figures, and improper statements in the revised paper.

---

> ### Author Response · Authors · 2025-11-24
>
> Dear Reviewer 4kU3,
>
> We greatly appreciate your time and effort in reviewing our work.
> **We are eager to ensure that we have adequately addressed your concerns and are prepared to offer further clarifications or address any additional questions you may have.**
> We would be grateful if you could share your thoughts on our rebuttal.
>
> Best regards,
>
> The Authors

---

> > ### Comment · Reviewer_4kU3 · 2025-11-26
> >
> > Thanks for the rebuttal, and most of my concerns are addressed.
> >
> > However, as a new task, the rationality or application is not discussed adequately yet.
> > Firstly, the one-shot task is compared, but the required one-shot seem not "large-scale".
> > Secondly, as stated that "in the inference phase, the reference images that two tasks input are the same". Therefore, the proposed method still requires one-shot, and thus the issue of annotation consumption still exists.
> > Therefore, I still concern on the necessity of the proposed task agasint one-shot setting.

---

> ### Author Response · Authors · 2025-11-26
>
> **Dear Reviewer 4kU3,**
>
> **Thank you for taking the time to provide additional feedback and share your concerns with us.**
> **We sincerely appreciate your comments and would like to address your points with the following clarifications.**
>
> >However, as a new task, the rationality or application is not discussed adequately yet. Firstly, the one-shot task is compared, but the required one-shot seem not "large-scale". Secondly, as stated that "in the inference phase, the reference images that two tasks input are the same". Therefore, the proposed method still requires one-shot, and thus the issue of annotation consumption still exists. Therefore, I still concern on the necessity of the proposed task agasint one-shot setting.
>
> We apologize for the confusion.
>
> (1) We deeply apologize for the confusion caused by the inappropriate phrasing "large-scale".
> RefCD aims to eliminating the detector’s reliance on manual annotations in the training stage, rather than highlighting data scale. In order to quantitatively evaluate the performance of RefCD, we compare it with several one-shot detectors. Both the compared one-shot detectors and our RefCD are not large-scale. The paper has been revised to avoid using the  term of 'large -scale'.
>
> (2) Due to the absence of category-aware unsupervised detectors, we follow the one-shot setting for comparisons to provide quantitative and fair comparison with other detectors on public benchmarks. For the evaluated open-set detection methods (*e.g.*, open vocabulary detection, reference-based detection, one-shot detection), they usually require a prompt for detection.
> Specifically, open vocabulary detectors use texts as prompts, while reference-based and one-shot methods use reference images as prompts. Detectors identify objects of interest based on these prompts during inference.
>
> Putting the prompts aside, these methods usually  have their own contributions in terms of training approaches or network architectures. As  for RefCD, it  aims to eliminate the detector’s reliance on manual annotations in  the training stage, rather than to distinguish itself from existing (one-shot) methods in the inference stage.

---

> > ### Comment · Reviewer_4kU3 · 2025-11-27
> >
> > My core concern is on the rationality or application of the task.
> >
> > I have questioned that "RefCD also uses reference images, but why is it without class label? What's the difference between the reference images in two tasks? " Only the difference on reference usages is explained. More simply, how to collect the reference images for RefCD? Because it is without class label, could we randomly sample images, e.g., OOD object images or just scene images, as the reference images for RefCD?
> >
> > From anothr perspective, In Figure 1 (c), the reference images in Unsupervised Training seem to be well box-cropped and from the desired or target categories. Correspondingly, it seems not in a strict Unsupervised Training, and also requires some annotations similar to one-shot method due to the well box-cropping for the Unsupervised Training. Therefore, I concern on the rationality or application of the proposed task against one-shot task.
> >
> > The answer could clarify how many annotations could be eliminated.

---

> > > ### Author Response · Authors · 2025-11-28
> > >
> > > >From anothr perspective, In Figure 1(c), the reference images in Unsupervised Training seem to be well box-cropped and from the desired or target categories. Correspondingly, it seems not in a strict Unsupervised Training, and also requires some annotations similar to one-shot method due to the well box-cropping for the Unsupervised Training. Therefore, I concern on the rationality or application of the proposed task against one-shot task. The answer could clarify how many annotations could be eliminated.
> > >
> > > The rationality/application of the proposed task against one-shot task is discussed to some extent in the above. Here, we will give some explanations.
> > >
> > > **(1) Box-cropped reference image.**
> > >
> > > Figure 1(c) presents diagrams of RefCD. In practice, the generated pseudo-boxes cannot accurately enclose objects with clear categorical meanings. Additional examples of pseudo-boxes are provided in Appendix A.6 Figure 12.
> > >
> > > The boxes we used are all pseudo boxes (rather than human-provided boxes), and no category information is used (even in the pseudo box generation procedure). The generation of pseudo boxes fully follows the unsupervised methods CutLER [1] (CVPR 2023) and TokenCut [2] (TPAMI 2023), which are well-defined and widely accepted in the unsupervised detection field (the details of the pseudo box generation are provided in Appendix A.6). From this perspective, our method is indeed an unsupervised method. As stated before, one-shot methods need manual annotations for training. Though the manual annotated boxes may be replaced by pseudo boxes, the category annotation is inevitable and irreplaceable for the training of one-shot methods. This is the rationality/application of the proposed task against one-shot task.
> > >
> > > **(2) How many annotations could be eliminated?**
> > > At the training stage, all (manual, more specifically) annotations of RefCD can be eliminated.
> > >
> > > At the inference stage, the reference image is provided by users, which means that a box may be required if the reference image is cropped from a larger image. As for the category label, RefCD does not explicitly require category labels. However, as stated above, the implicit category grounding exists in all detectors as long as the detectors are used by users to detect some specific objects.
> > >
> > > [1] Wang, Yangtao, et al. "Tokencut: Segmenting objects in images and videos with self-supervised transformer and normalized cut." IEEE transactions on pattern analysis and machine intelligence 45.12 (2023): 15790-15801.
> > > [2] Wang, Xudong, et al. "Cut and learn for unsupervised object detection and instance segmentation." Proceedings of the IEEE/CVF conference on computer vision and pattern recognition. 2023.

---

> > > > ### Comment · Reviewer_4kU3 · 2025-11-28
> > > >
> > > > An eligible paper should detailedly and explicitly list the settings of training and inference, which are missing in the original submission. After the rebuttal, my concerns are well addressed. Overall, the task is practical, and the method is adequate. Therefore, I will update my ratings if feasible (the edit button is invisible at this time).

---

> ### Author Response · Authors · 2025-11-28
>
> **Dear Reviewer 4kU3,**
>
> **Your thoughtful consideration of our responses is sincerely appreciated. Engaging in this discussion with you has been truly rewarding.**
>
> >I have questioned that "RefCD also uses reference images, but why is it without class label? What's the difference between the reference images in two tasks? " Only the difference on reference usages is explained. More simply, how to collect the reference images for RefCD? Because it is without class label, could we randomly sample images, e.g., OOD object images or just scene images, as the reference images for RefCD?
>
> To better understand the relation between the reference image and the class label, and the collection of reference images in RefCD, we should know the training and inference details.
>
> In the training stage, we follow the unsupervised methods CutLER [1] (CVPR 2023) and TokenCut [2] (TPAMI 2023) to train our model on the images from ImageNet without using the class annotations (only the generated pseudo boxes are used). And the reference image is cropped from randomly selected images based on the corresponding pseudo boxes. It is clear to see that reference images are without class labels in RefCD in the training stage.
> However, for the compared one-shot methods, they all need the class labels for training. To eliminate the reliance on expensive manual annotations (eg., class labels), our method is more necessary, more applicable, and more rational than the compared one-shot methods.
> This is the key difference between RefCD and one-shot methods.
>
> In the inference stage, the reference image should usually be provided by users, and the objects that are interested by users are indicated by the reference image. There is no explicit class label with the reference image. However, we admit that the concept of class is implicitly grounded in the reference image by users. However, the implicit class grounding exists in all detectors (no matter whether supervised or unsupervised detectors) as long as the detectors are used by users to detect some specific objects.
> But differently, the collection of the reference image or RefCD is more flexible, and OOD object images and some scene images also work. The reason is that RefCD learns the similarity between objects and the reference image. Visualization results for several specialized reference images (e.g., legs of person, wave, and people skiing on the snow) are provided in Appendix A.1 and Figure 8. In addition, in our **Author’s Rebuttal 1** to **Reviewer VNJE**, RefCD also shows great generalization capability to some specific domains.
> Overall, our method is more general and more applicable than the compared one-shot methods.
> This is another difference between RefCD and existing one-shot methods

---

> ### Author Response · Authors · 2025-11-28
>
> Dear Reviewer 4kU3,
>
> We sincerely appreciate your thoughtful consideration of our rebuttal. It has been a pleasure engaging in this discussion with you, and your insightful comment have helped improve the inadequacies in our paper. Implement details in Section 4.1 of the paper has been updated.
>
> Best regards,
>
> The Authors

---

### Official Review · Reviewer_VNJE · 2025-10-30

**Soundness:** 3
**Presentation:** 3
**Contribution:** 2
**Rating:** 4
**Confidence:** 4

**Summary:**

This paper introduces an unsupervised reference-based object detection framework, RefCD, designed to enable open-set object detection without dependence on human annotations. The proposed method first performs category-agnostic detection to localize potential objects in an image, followed by category-aware matching that leverages feature similarity between reference images and target objects to identify instances of specific categories. A central contribution of this work is the introduction of a Feature Similarity Loss (FS Loss), which encourages the model to capture latent category structures among objects during unsupervised training. Extensive experiments on multiple benchmark datasets indicate that RefCD attains competitive performance in unsupervised object detection and achieves results comparable to several supervised approaches, demonstrating its robustness and potential applicability in annotation-free settings.

**Strengths:**

* This work proposes a concise and well-structured pipeline that integrates self-supervised representations, Hungarian assignment, and similarity-based objectives. Each component is reasonably motivated and supported by empirical evidence, contributing to the coherence and effectiveness of the overall approach.
* The presentation of the study is clear and systematic, with a transparent problem formulation, well-defined objectives, and detailed optimization and implementation procedures. The descriptions of pseudo-box generation and reference encoding are particularly helpful, and the accompanying figures and qualitative analyses enhance the interpretability of the results.
* The experimental evaluation is thorough, covering a range of datasets and task settings, and includes comparisons with both unsupervised and supervised baselines. The analysis offers useful observations regarding loss variants, similarity measures, and Top-K selection, and also includes an examination of challenging cases such as crowded or occluded scenes.
* The work provides meaningful evidence that label-free, open-set, category-aware detection is feasible and, in certain cases, can approach the performance of supervised methods. This finding highlights the potential for flexible category specification without retraining and suggests a possible path toward reducing annotation costs in Open Vocabulary Object Detection.

**Weaknesses:**

* The model’s performance appears to rely heavily on the semantic richness of the DINOv2 ViT features (Table 6). In domain-specific scenarios where such pretraining is unavailable or less effective, the generalization and stability of RefCD remain uncertain.
* The study employs four sets of reference images for evaluation (Appendix A.7). However, other works may adopt different template sets. Without a publicly shared reference pool and accompanying scripts, direct cross-paper comparison of quantitative results is difficult to ensure.
* Table 11 combines results from fully unsupervised training (using pseudo boxes) and box-only, label-free training on COCO BASE, which in practice constitutes weak supervision. Clarifying this distinction would help prevent potential overstatement of the unsupervised setting.
* The category-aware detection results are influenced by several hyperparameters—thresholds, the sigmoid temperature (set to 10), and the choice of top-K queries (Table 3; Fig. 4). A more systematic analysis would provide deeper insight into the robustness of these design choices.
* While Fig. 5 demonstrates qualitative success in extending RefCD to single-object tracking, quantitative results on standard tracking benchmarks are absent, which limits the strength of this extension claim.

**Questions:**

* Beyond the configurations reported in Table 7, have the authors conducted a broader sweep over $\alpha \in {1,\dots,4}$ and $\beta \in {3,\dots,7}$? Additionally, how sensitive is the model to the fixed temperature parameter (set to $10$), and how does it interact with the similarity threshold used for category-aware matching?
* Instead of selecting a single strongest positive per reference, have the authors explored treating queries with high pseudo similarity $p_n > \tau$ as soft positives, or directly regressing $\hat{p}_n$ toward $p_n$? It would be helpful to know whether such strategies yield measurable improvements in recall or category coverage.
* Could the authors provide an ablation or visualization illustrating the impact of the Top-$K$ value (Table 3) and discuss robustness to batch size, as well as the calibration of the foreground confidence head that determines the Top-$K$ queries?
* Appendix A.7 indicates that the reference templates include both COCO and ImageNet images. Are the comparisons with SINE and UNICL-SAM conducted on exactly the same reference set? If not, could the authors include a sensitivity analysis to quantify the effect of template variation?
* In Table 11, the “COCO BASE w/o category labels” configuration effectively constitutes box-only (weakly supervised) training. It would be beneficial to clarify this setup in the main text and report its results separately from those of the fully unsupervised setting.
* Figure 4 suggests that adjusting the similarity threshold enables finer-grained subclass separation (e.g., dog breeds). Would employing a multi-threshold or re-calibration mechanism further mitigate subclass confusion or reduce false positives at inference?
* For the crowded, composite, or occluded scenes discussed in Appendix A.4, have the authors considered incorporating mask-level similarity or iterative box-refinement mechanisms to improve localization? If so, any preliminary findings would be valuable to include.

---

> ### Author Response · Authors · 2025-11-20
> **Authors' Rebuttal 1**
>
> **Dear Reviewer VNJE:**
>
> **We greatly appreciate your detailed feedback and recognition of our work. We provide more detailed experiments and analyses below. We also include exploratory experiments on RefCD. We hope our response will address your questions and improve the score of our work.**
>
> **W1: Generalization and stability of RefCD**
>
> >The model’s performance appears to rely heavily on the semantic richness of the DINOv2 ViT features (Table 6). In domain-specific scenarios where such pretraining is unavailable or less effective, the generalization and stability of RefCD remain uncertain.
>
> We sincerely thank the reviewer for raising this critical question. Undeniably, the semantic richness of DINOv2 contributes to improving RefCD’s detection performance.
>
> However, RefCD achieves category-aware detection based on feature similarity between objects.
> It does not predict specific category labels using category features.
> Thus, domain-specific scenarios do not affect generalization of RefCD.
> To validate this, we have supplemented experiments on domain-specific scenarios in the Appendix A.9 and Figure 12.
> We report experimental results on industrial and underwater scenarios.
> The industrial scenario is selected from the [RAD dataset](https://github.com/hustCYQ/RAD-dataset) [1], comprising 4 categories and 1224 industrial scene images.
> The underwater scenario is selected from the [3D-ZeF20 dataset](https://motchallenge.net/data/3D-ZeF20/) [2], with 'zebra fish' as the objects of interest. Since the test set does not provide ground-truth, the training set is used for one-shot evaluation.
>
> As shown in R-Table 2, RefCD still enables effective detection in domain-specific scenarios. Although the scenes of RAD and 3D-ZeF20 are highly specialized, the object complexity within these scenes is lower than that of COCO NOVEL, so that RefCD achieves even better performance in these scenarios.
>
> Notably, RefCD is pre-trained on ImageNet with a frozen reference encoder, without fine-tuning on RAD or 3D-ZeF20. This demonstrates the generalization capbility of RefCD to domain-specific scenarios.
>
>
> ### R-Table 2: Results in domain-specific scenarios.
>
> | Scenarios | Datasets     | AP₅₀ | AP   | AR   |
> | :-------: | :----------: | :--: | :--: | :--: |
> | industry  | RAD [1]      | 44.3 | 30.7 | 57.2 |
> | underwater| 3D-Zef20 [2] | 50.0 | 28.1 | 61.0 |
>
>
> [1] Cheng, Yuqi, et al. "Rad: A comprehensive dataset for benchmarking the robustness of image anomaly detection." 2024 IEEE 20th International Conference on Automation Science and Engineering (CASE). IEEE, 2024.
> [2] Pedersen, Malte, et al. "3d-zef: A 3d zebrafish tracking benchmark dataset." Proceedings of the IEEE/CVF Conference on Computer Vision and Pattern Recognition. 2020.
>
> **W2 & Q4: Fairness and reproducibility of experiments**
>
> >W2: The study employs four sets of reference images for evaluation (Appendix A.7). However, other works may adopt different template sets. Without a publicly shared reference pool and accompanying scripts, direct cross-paper comparison of quantitative results is difficult to ensure.
>
> >Q4: Appendix A.7 indicates that the reference templates include both COCO and ImageNet images. Are the comparisons with SINE and UNICL-SAM conducted on exactly the same reference set? If not, could the authors include a sensitivity analysis to quantify the effect of template variation?
>
> Thank the reviewer for noticing the experimental details.
>
> Since SINE has not made publicly available the reference images used on COCO NOVEL, UNICL-SAM did not report detection evaluations on COCO NOVEL in their paper. Therefore, we conducted a fair comparison among RefCD, SINE and UNICL-SAM using the four reference image sets specified in Appendix A.7. The results are presented in Table 1 of the main paper.
>
> To facilitate direct cross-paper comparison and result reproduction, we will publicly release relevant code and data, including the generated pseudo-box annotation JSON files, selected reference images, and scripts for random reference image selection.

---

> ### Author Response · Authors · 2025-11-20
> **Authors' Rebuttal 2**
>
> **W3 & Q5: Clarification of unsupervised and weakly supervised training**
>
> >W3: Table 11 combines results from fully unsupervised training (using pseudo boxes) and box-only, label-free training on COCO BASE, which in practice constitutes weak supervision. Clarifying this distinction would help prevent potential overstatement of the unsupervised setting.
>
> >Q5: In Table 11, the “COCO BASE w/o category labels” configuration effectively constitutes box-only (weakly supervised) training. It would be beneficial to clarify this setup in the main text and report its results separately from those of the fully unsupervised setting.
>
> We apologize for the confusion.
>
> We have clarified the purpose of weak supervision training and its experiment setting in Section 4.5 and Appendix A.3 of the paper.
> We have also separated the experimental results of weakly supervision and fully unsupervised learning into different subsections and tables. The experimental results of weak supervision training RefCD is shown in R-Table 9 (Table 14 in Appendix).
> Detailed experiments and analyses are provided in Appendix A.3.3 and answer for Question 7.
>
>
>
> **W4 & Q1 & Q3: More ablation experiments and analyses**
>
> >W4: The category-aware detection results are influenced by several hyperparameters—thresholds, the sigmoid temperature (set to 10), and the choice of top-K queries (Table 3; Fig. 4). A more systematic analysis would provide deeper insight into the robustness of these design choices.
>
> >Q1: Beyond the configurations reported in Table 7, have the authors conducted a broader sweep over $\alpha \in {1，\dots，4}$ and $\beta \in {3，\dots，7}$ ? Additionally, how sensitive is the model to the fixed temperature parameter (set to 10), and how does it interact with the similarity threshold used for category-aware matching?
>
> **(1) Results of different $\alpha$ and $\beta$.**
>
> Thank you for the insightful comments.
> More extensive ablation experiments on the two hyperparameters $\alpha \in {1，\dots，4}$ and $\beta \in {3，\dots，7}$ are conducted in the main paper.
> The results are presented in R-Table 3, and the Table 7 in the main paper also has been updated.
> Within this wider range of values for $\alpha$ and $\beta$, the combination of $\alpha=2$ and $\beta=4$ remains the optimal set of hyperparameters.
>
> ### R-Table 3: Results of feature similarity loss with different α and β
>
> | α   | β   | AP₅₀ | AP   | AR   |
> | :--: | :--: | :--:  | :--:  | :--:  |
> | 1   | 3   | 23.5 | 14.7 | 28.2 |
> | 1   | 4   | 22.1 | 12.9 | 23.3 |
> | 2   | 4   | **24.7** | **15.1** | **29.3** |
> | 2   | 5   | 24.0 | 14.9 | 27.7 |
> | 3   | 5   | 23.8 | 15.0 | 28.9 |
> | 3   | 6   | 23.7 | 14.7 | 28.7 |
> | 4   | 6   | 23.5 | 14.7 | 28.8 |
> | 4   | 7   | 22.3 | 13.9 | 28.0 |
>
> **(2) Results of different temperature parameter.**
>
> Thank you for the insightful comments.
> Ablation experiments and analyses on sensitivity of RefCD to the temperature parameter are supplemented, with results presented in R-Table 4.
> Corresponding Section 4.4 and Table 8 have also added in the main paper
>
>
> The temperature parameter serves to adjust the standard deviation between feature similarities.
> When temperature ≤ 1, the feature similarity between positive and negative samples is very close, hindering the model from learning accurate category-aware features.
> When the temperature is excessively large (*e.g.*, temperature ≥ 100), the predicted feature similarities tend to approach 0 or 1 after activation.
> This can lead to the neglect of some objects within the same class with lower similarity, reducing the detector’s generalization to objects of the same category.
>
> In the training and inference phase, the temperature parameter is only used as a parameter in activation function during category-aware matching and does not interact with the similarity threshold.
> Our goal is to ensure that objects of different categories exhibit obvious differences in feature similarity.
> A small temperature requires careful selection of a similarity threshold for each reference image.
> While an large temperature renders this threshold ineffective, as inevitable minor feature differences exist between intra-class objects.
>
> Thus, when the temperature set to 10, RefCD can well adapt to a universal similarity threshold and achieve the best result (2-nd row in R-Table 4).
>
> ### R-Table 4: Results of different temperature parameter.
> | Temperature | AP₅₀ | AP   | AR   |
> | :---------: | :--: | :--: | :--: |
> | 1           | 22.3 | 13.7 | 26.8 |
> | 10          | **24.7** | **15.1** | **29.3** |
> | 100         | 15.3 | 8.9  | 18.1 |

---

> ### Author Response · Authors · 2025-11-20
> **Authors' Rebuttal 3**
>
> >Q3: Could the authors provide an ablation or visualization illustrating the impact of the Top-K value (Table 3) and discuss robustness to batch size, as well as the calibration of the foreground confidence head that determines the Top-K queries?
>
> Thank you for the insightful comments.
>
> Ablation results on Top-K are reported in R-Table 5, and Table 3 in the main paper has been updated.
> It can be observed that RefCD achieves the best result on COCO NOVEL when K is set to 100.
> Note that Top-K refers to selecting the K queries for objects in each image to calculate the feature similarity loss.
> For each batch, each image within the batch selects K queries to compute the FS loss.
> This results in a total of batch size × K queries used for loss calculation.
> Thus, the parameter K is independent of the batch size.
>
> ### R-Table 5: Results of different Top-K value.
> | Top-K |  | COCO val2017 |  |  | COCO NOVEL |  |
> | :---: | :---: | :---: | :---: | :--: | :---: | :---: |
> |       | AP   | AP₅₀  | AR    | AP   | AP₅₀  | AR   |
> | 50    | 12.4 | 23.2  | 33.0  | 14.9 | 24.4  | 29.1 |
> | 100   | **12.9** | **23.6**  | **33.9**  | **15.1** | **24.7**  | **29.3** |
> | 200   | 12.8 | 23.4  | 33.3  | 15.0 | 24.4  | 29.0 |
> | 300   | 12.5 | 23.2  | 32.8  | 15.0 | 24.3  | 28.7 |
>
> **W5: Quantitative comparison of SOT task**
>
> >While Fig. 5 demonstrates qualitative success in extending RefCD to single-object tracking, quantitative results on standard tracking benchmarks are absent, which limits the strength of this extension claim.
>
> Thank you for the insightful comments.
>
> We report the quantitative results of RefCD on the unsupervised single object tracking (SOT) task, as presented in R-Table 6 (Table 9 of main paper).
> We conducted the SOT task evaluation on the VOT 2018 [1] dataset.
> It can be observed that RefCD achieves comparable performance on the SOT task compared to traditional method KCF[2] and the best unsupervised SOT method.
>
> ### R-Table 6: Results of single object tracking.
> | Method        | A ↑   | R ↓   | EAO ↑ |
> | :-----        | :---: | :---: | :---: |
> | KCF [2]       | 0.447 | 0.773 | 0.135 |
> | USOT [3]      | 0.564 | 0.435 | 0.290 |
> | RefCD(Ours)   | 0.511 | 0.577 | 0.187 |
>
> [1] Kristan, Matej, et al. "The sixth visual object tracking vot2018 challenge results." Proceedings of the European Conference on Computer Vision (ECCV) workshops. 2018.
> [2] Henriques, João F., et al. "High-speed tracking with kernelized correlation filters." IEEE Transactions on Pattern Analysis and Machine Intelligence 37.3 (2014): 583-596.
> [3] Zheng, Jilai, et al. "Learning to track objects from unlabeled videos." Proceedings of the IEEE International Conference on Computer Vision. 2021.
>
> **Q2: Exploration of positive samples**
>
> >Instead of selecting a single strongest positive per reference, have the authors explored treating queries with high pseudo similarity $p_n > \tau$ as soft positives, or directly regressing $\hat{p}_n$ toward $p_n$? It would be helpful to know whether such strategies yield measurable improvements in recall or category coverage.
>
> We appreciate the reviewer’s insightful comment on theoretical extensions.  Different positive sample selection strategies do affect recall or category coverage, and we have explored related methods in RefCD.
>
> We initially set a soft threshold $\tau=0.6$ for positive samples, treating samples with $p_n > \tau$ as objects of interest. However, as shown in the 2-nd row of Table 7, the detection performance is suboptimal.
> We attribute this to potential intra-class variations among objects of the same category: using the soft threshold $\tau$ may exclude potential objects of interest, leading to incorrect loss calculation and preventing the model from learning category-aware features.
>
> Consequently, we adopt a direct approach of regressing $\hat{p}_n$ toward $p_n$.
> Experimental validation show that allowing the model to autonomously learn feature similarities yields improvements in both recall and category coverage compared to using the soft threshold $\tau$.
>
> Following Reviewer VNJE’s suggestion, we conducted more detailed ablation experiments, as presented in R-Table 7.
> The results show that lower values of $\tau$ lead to improvements in detection performance. When $\tau=0.9$, RefCD can only detect objects that are completely consistent with the reference images. When $\tau=0$ (*i.e.*, directly regressing $\hat{p}_n$ toward $p_n$, RefCD achieves the best performance on COCO NOVEL.
>
> The corresponding experiments and analyses have been added in Section 4.4 and Table 9 of the main paper.
>
>
> ### R-Table 7: Results of different τ.
> | Positive Sample | AP₅₀ | AP   | AR   |
> | :-------------: | :--: | :--: | :--: |
> | τ=0.9           | 13.4 | 8.0  | 15.7 |
> | τ=0.6           | 18.7 | 11.4 | 20.4 |
> | τ=0.3           | 22.8 | 13.2 | 27.1 |
> | τ=0 (regressing $\hat{p}_n$ toward $p_n$) | **24.7** | **15.1** | **29.3** |

---

> ### Author Response · Authors · 2025-11-20
> **Authors' Rebuttal 4**
>
> **Q6: Finer-grained subclass separation**
>
> >Figure 4 suggests that adjusting the similarity threshold enables finer-grained subclass separation (e.g., dog breeds). Would employing a multi-threshold or re-calibration mechanism further mitigate subclass confusion or reduce false positives at inference?
>
> We appreciate the reviewer’s insightful comment on theoretical extensions.
>
> Multi-thresholds do mitigate subclass confusion and reduce false positives at inference, but they increase the time cost of threshold selection.
> To demonstrate the effectiveness of multi-thresholds while avoiding excessive workload, we selected a subset of data for testing.
> Specifically, we chose 50 dog-containing images from COCO val2017 to test RefCD.
> We divided them into 10 categories based on dog breeds, including *Husky*, *Doberman*, and *Chinese Rural Dog* *etc.*, and recalibrated the multi-thresholds for each breed.
> The experimental results are shown in 2-nd row of Table 8.
> Compared with using a single universal threshold, the multi-threshold and re-calibration mechanism further alleviated subclass confusion and reduced false positives during inference.
>
> Please note that **for fair comparisons with other methods, we set a single threshold that shared by all categories during evaluation**.
>
> ### R-Table 8: Results of multi-threshold for different category.
> | Method           | AP₅₀ | AP   | AR   |
> | :--------------  | :--: | :--: | :--: |
> | Single-threshold | 34.3 | 22.1 | 39.1 |
> | Multi-threshold  | 40.2 | 29.7 | 41.3 |
>
> **Q7: Exploring solutions for crowded scenarios**
>
> >For the crowded, composite, or occluded scenes discussed in Appendix A.4, have the authors considered incorporating mask-level similarity or iterative box-refinement mechanisms to improve localization? If so, any preliminary findings would be valuable to include.
>
> We appreciate the reviewer’s insightful comment on theoretical extensions.
> In response, we report experiments and analyses using mask-level similarity and iterative box-refinement mechanisms.
> Corresponding results are added in Appendix A.3 of the paper. (To improve the logical coherence of the appendix, we have relocated the original Section A.4 to Section A.1.)
>
> **(1) Mask-level similarity.**
>
> Based on your feedback, we have explored the mask-level feature similarity to improve localization.
> Specifically, we use mask labels from COCO BASE to train RefCD under weak supervision.
> As shown in the 1-st row of R-Table 9, RefCD trained with mask-level feature similarity outperforms that with box-level feature similarity.
> We agree that mask-level features focus more on the object itself.
> They reduce feature interference from the background and other objects, leading to more accurate detection results.
> We add relevant visualization results in Appendix A.3 Figure 10.
> It shows that mask-level feature similarity improves detection results in occluded scenes.
>
> **(2) Box-refinement mechanisms.**
>
> RefCD uses multi-layer decoders to improve localization.
> We use 6 Transformer layers as decoder in RefCD.
> We conduct experiments on RefCD (weakly supervised trained on COCO BASE) and provide results for each decoder layer.
> As shown in last 6 rows in R-Table 9, iterative box-refinement mechanisms help improve detection performance. The 6-th decoder layer outputs the best detection results.
>
>
> ### R-Table 9: Results of weakly supervised training RefCD.
> | Method       | Paradigm          | Dataset  | Feature Level | Output Layer | AP_box |
> | :----------: | :---------------: | :------: | :-----------: | :----------: | :----: |
> | RefCD(ours)  | Weakly Supervised | COCO BASE| Mask-level    | 6            | 28.3   |
> | RefCD(ours)  | Weakly Supervised | COCO BASE| Box-level     | 6            | 24.9   |
> | RefCD(ours)  | Weakly Supervised | COCO BASE| Box-level     | 5            | 24.5   |
> | RefCD(ours)  | Weakly Supervised | COCO BASE| Box-level     | 4            | 24.0   |
> | RefCD(ours)  | Weakly Supervised | COCO BASE| Box-level     | 3            | 23.3   |
> | RefCD(ours)  | Weakly Supervised | COCO BASE| Box-level     | 2            | 22.2   |
> | RefCD(ours)  | Weakly Supervised | COCO BASE| Box-level     | 1            | 19.0   |

---

> ### Author Response · Authors · 2025-11-24
>
> Dear Reviewer VNJE,
>
> We greatly appreciate your time and effort in reviewing our work.
> **We are eager to ensure that we have adequately addressed your concerns and are prepared to offer further clarifications or address any additional questions you may have.**
> We would be grateful if you could share your thoughts on our rebuttal.
>
> Best regards,
>
> The Authors

---

> ### Author Response · Authors · 2025-11-28
>
> Dear Reviewer VNJE,
>
> We hope this message finds you well. We apologize for any inconvenience caused by reaching out over the weekend. We truly appreciate your engagement with our rebuttal and thank you for your insightful comments and questions regarding detailed experiments.
>
> As the rebuttal discussion period ends in 5 days, we would be grateful for your feedback on whether our responses have adequately addressed your concerns. We are ready to answer any further questions you may have.
>
> Thank you for your valuable time and effort!
>
> Best regards,
>
> The Authors

---

### Official Review · Reviewer_DSqC · 2025-11-01

**Soundness:** 3
**Presentation:** 3
**Contribution:** 2
**Rating:** 4
**Confidence:** 4

**Summary:**

This paper addresses the challenge of unsupervised object detection, where the goal is to detect objects without access to annotated category labels. It introduces RefCD (Reference-based Category Discovery), a novel framework inspired by supervised one-shot object detection methods. Similar to one-shot paradigms—which use a reference image to localize instances of the same class—RefCD leverages feature similarity between a reference image and candidate regions to perform category-aware object detection in an unsupervised setting.

**Strengths:**

The paper proposes a novel perspective on unsupervised object detection by combining reference images with feature-similarity-based matching, opening a new direction in this research area.

Extensive experiments show strong performance, with RefCD outperforming existing methods in unsupervised object detection benchmarks.

**Weaknesses:**

The paper attempts to differentiate its approach from one-shot object detection. However, the objective of the proposed method is essentially the same as one-shot object detection; the primary distinction lies in the training procedure. While the work present a new problem setting that they argue differs from existing detection paradigms, the main distinction appears to be the replacement of human annotation with unsupervised detection method, such as CutLER. This reduces the strength of the contribution.

Since the reference images are unlabeled, it is unclear how the method distinguishes between different object categories, i.e., how one can determine whether two reference images correspond to different classes. Additionally, if a reference image contains only a partial view of an object (e.g., just the wheels of a car), it is unclear whether the model should detect the full object (the car) or only the referenced part (the wheels). This ambiguity raises questions about how RefCD handles partial-object references and category granularity.

**Questions:**

In Figure 1, if we rename “deer” as ref1 and “capybaras” as ref2, what is the difference in the detection objective between (a) and (c)?

---

> ### Author Response · Authors · 2025-11-20
> **Authors' Rebuttal**
>
> **Dear Reviewer DSqC:**
>
> **We greatly appreciate your detailed feedback and recognition of our work. We hope the following response will address your questions and improve the score of our work.**
>
> **W1: Differences from previous works.**
>
> >The paper attempts to differentiate its approach from one-shot object detection. However, the objective of the proposed method is essentially the same as one-shot object detection; the primary distinction lies in the training procedure. While the work present a new problem setting that they argue differs from existing detection paradigms, the main distinction appears to be the replacement of human annotation with unsupervised detection method, such as CutLER. This reduces the strength of the contribution.
>
> We apologize for the confusion.
>
> Our RefCD focuses on unsupervised object detection, with one-shot methods only for quantitative comparison.
> Existing one-shot detectors heavily rely on manual category labels and are incompatible with unsupervised settings, while unsupervised methods fail to learn category information.
>
> To address this gap, RefCD introduces unlabeled reference images as semantic prompts and designs a Feature Similarity (FS) loss to enable category-aware learning from unlabeled pseudo-boxes.
> Unlike one-shot detection, RefCD does not require predefined categories, only unlabeled reference images are used to detect objects of interest.
> Due to the lack of unsupervised baselines for category-aware detection, we adopt supervised one-shot experimental settings for comparison.
> As shown in Table 2, unsupervised RefCD outperforms most supervised one-shot methods.
>
> We sincerely thank you for your suggestions. We have updated the paper (especially Section 1 Introduction) to clarify these points.
>
>
> **W2: Ambiguous category definition.**
>
> >Since the reference images are unlabeled, it is unclear how the method distinguishes between different object categories, i.e., how one can determine whether two reference images correspond to different classes. Additionally, if a reference image contains only a partial view of an object (e.g., just the wheels of a car), it is unclear whether the model should detect the full object (the car) or only the referenced part (the wheels). This ambiguity raises questions about how RefCD handles partial-object references and category granularity.
>
> We apologize for the confusion.
>
> (1) RefCD does not perform explicit category definition.
> If multiple reference images are provided simultaneously, they are treated as distinct categories.
> Detected objects are mutually exclusively assigned to the reference image with the highest feature similarity.
>
> For example, inputting both a *Husky* and a *Corgi* as reference images in one inference run implicitly defines two distinct categories (*i.e.*, dog breeds).
> *Husky* and *Corgi* in the target image will be matched to their corresponding reference images.
> In contrast, when only a *Husky* (or *Corgi*) reference image is used, detection results are determined by the similarity threshold: a low threshold identifies both *Husky* and *Corgi* as objects of interest, while a higher threshold only detects *Husky* (or *Corgi*).
> (**Note that** *Husky* **and** *Corgi* **represent the corresponding images/objects, not explicit category definition.**)
>
> **For unified quantitative evaluation, we define the grounded categories of reference images during the evaluation process. This enables comparison with other methods.**
>
> (2) Our detector faithfully detects objects similar to the reference images.
> Relevant visualization results are added in the Appendix A.1 (Figure 8).
> Detection results depend on reference images.
> If the reference image is a partial region of an object (*e.g.*, a wheel), the detected objects will correspond to that target region (*e.g.*, the wheel part).
> Note that during quantitative evaluation, we use complete reference images of the corresponding categories for evluation.
>
> **Q1: Differences with previous one-shot detection methods**
>
> >In Figure 1, if we rename “deer” as ref1 and “capybaras” as ref2, what is the difference in the detection objective between (a) and (c)?
>
> We apologize for the confusion. Figure 1 has been updated in the paper.
>
> Figure 1 presents diagrams of different detection methods.
> In Figure 1(a) in submitted version, "deer" and "capybaras" represent explicit category definitions.
> In Figure 1(c) in submitted version, "ref1" and "ref2" are used for visualization, rather than the category label defination. They are not category labels.
> RefCD is an unsupervised detector that does not require input category labels.
>
> To avoid confusion, Figure 1 has been updated in the rebuttal version. We remove the textual explanation (*i.e.*, "ref1" and "ref2") and employ colors (red and green) to distinguish different reference images.

---

> ### Author Response · Authors · 2025-11-24
>
> Dear Reviewer DSqC,
>
> We greatly appreciate your time and effort in reviewing our work.
> **We are eager to ensure that we have adequately addressed your concerns and are prepared to offer further clarifications or address any additional questions you may have.**
> We would be grateful if you could share your thoughts on our rebuttal.
>
> Best regards,
>
> The Authors

---

> > ### Comment · Reviewer_DSqC · 2025-11-26
> >
> > Thank you for the detailed response. The definition and organization of categories used in this work are not clearly explained. For example, ImageNet adopts WordNet to construct and structure semantic categories, providing a well-defined hierarchy and disambiguation between classes. By contrast, it remains unclear how this work defines, groups, or differentiates categories: whether they are derived from an existing ontology, manually curated, or constructed according to some specific criteria. A more precise and transparent description of the category definitions and their underlying rationale would greatly improve the clarity and reproducibility of the paper. For example, whether a Husky and a Corgi would be considered part of the same category or treated as distinct categories under the authors’ framework is not clear.

---

> > > ### Author Response · Authors · 2025-11-26
> > >
> > > **Dear Reviewer DSqC,**
> > >
> > > **Thank you for taking the time to provide additional feedback and share your concerns with us.**
> > >
> > > >The definition and organization of categories used in this work are not clearly explained. For example, ImageNet adopts WordNet to construct and structure semantic categories, providing a well-defined hierarchy and disambiguation between classes. By contrast, it remains unclear how this work defines, groups, or differentiates categories: whether they are derived from an existing ontology, manually curated, or constructed according to some specific criteria. A more precise and transparent description of the category definitions and their underlying rationale would greatly improve the clarity and reproducibility of the paper. For example, whether a Husky and a Corgi would be considered part of the same category or treated as distinct categories under the authors’ framework is not clear.
> > >
> > > We apologize for the confusion and appreciate your comments sincerely. Here we would like to clarify this concern.
> > >
> > > We do not involve category definitions during either training or inference, as we focus on the learning of feature similarity between objects and reference images. And RefCD does not need to explicitly predict category labels in both training and inference stages.
> > >
> > > During training, RefCD is trained on the images
> > > from ImageNet without  using the category  annotations (only the generated pseudo boxes are used).
> > >
> > > During inference, the objects in the  target  image that share high similarities with the reference image are detected. However, in order to quantitatively evaluate the performance of  RefCD, we follow existing works to compare different methods on COCO, where the category definitions of COCO are  adopted (just for  evaluation). Under the category definition of COCO, Huskies and Corgis are considered part of the same category ("dog").

---

### Official Review · Reviewer_eSLo · 2025-11-03

**Soundness:** 3
**Presentation:** 3
**Contribution:** 1
**Rating:** 4
**Confidence:** 4

**Summary:**

The paper proposes a method for unsupervised, reference-based detection and category discovery by assuming that similarity in feature space corresponds to implicit category membership, applying a relatively simple architecture and loss formulation to leverage that assumption, and demonstrating competitive empirical results.

**Strengths:**

The experimental analysis is good and extensive

**Weaknesses:**

The paper assumes that feature similarity corresponds to implicit category membership, intuitively plausible,  but how does the method cope in cases of high intra-class variation? For example, if in a scene there are red apples and green apples, and the reference annotation is on red apples, can the method reliably detect green apples of the same “apple” category?

Although the motivation is strong and the experiments fairly extensive, the architecture and loss design remain relatively straightforward, which suggests the contribution may lie more in the integration of existing modules rather than in fundamentally new methodology. Can the authors clarify which parts of the method are novel beyond existing unsupervised object-discovery and detection frameworks?

Specifically, there already exists a line of work exploring exemplar-based or reference-guided detection and recognition across various domains—including detection, counting, and segmentation. In the detection community specifically, methods such as  [1-3] have also utilized exemplar or reference inputs to guide detection, though typically in supervised or semi-supervised settings. In parallel, several text-driven approaches, for example  [4-5], conditioning detectors on textual prompts rather than visual exemplars.

[1]Large-Scale Unsupervised Object Discovery,
[2]OS2D: One-Stage One-Shot Object Detection,
[3]Siamese DETR
[4]OWL-ViT: Simple Open-Vocabulary Detection with Vision-Language Models,
[5]RegionCLIP: Region-based Language-Image Pretraining

**Questions:**

see Weaknesses

---

> ### Author Response · Authors · 2025-11-20
> **Authors' Rebuttal 1**
>
> **Dear Reviewer eSLo:**
> We greatly appreciate your detailed feedback and the time you spent reviewing our work. **We believe there may have been some misinterpretations of key elements, partially due to our inadvertent errors, which could influence your assessment.** We sincerely hope our clarifications will prompt you to re-review and re-evaluate our work. Thank you for your understanding and consideration.
>
> **W1: Cases with large intraclass differences**
>
> >The paper assumes that feature similarity corresponds to implicit category membership, intuitively plausible, but how does the method cope in cases of high intra-class variation? For example, if in a scene there are red apples and green apples, and the reference annotation is on red apples, can the method reliably detect green apples of the same “apple” category?
>
> Thank you for your insightful comments.
> As you have pointed out, intra-class variation do exist. But RefCD demonstrates robustness to intra-class variations.
> We have supplemented the corresponding visualization results in Figure 4 of the main paper.
>
> As illustrated in Figure 4, all accurate detection results exhibit high confidence, while finer-grained confidence intervals allow the model to distinguish between different dog breeds and apple color variants. Specifically, when an image contains both red and green apples and the reference annotation designates red apples, the model still categorizes green apples into the "apple" category. Owing to the color variation, the detection confidence of green apples is slightly lower than that of red apples, but this does not compromise the model’s accurate detection performance.
>
> **W2: Differences with existing unsupervised object-discovery and detection frameworks**
>
> >Although the motivation is strong and the experiments fairly extensive, the architecture and loss design remain relatively straightforward, which suggests the contribution may lie more in the integration of existing modules rather than in fundamentally new methodology. Can the authors clarify which parts of the method are novel beyond existing unsupervised object-discovery and detection frameworks?
>
> We sincerely apologize for the confusion.
> We elaborate on the contributions of our work and its distinctions from existing methods as follows:
>
> (1) **Our RefCD unsupervised learns category-aware information without any category labels, which is completely different from all existing methods.**
> As mentioned by **Reviewer DSqC**, "*the paper proposes a novel perspective on unsupervised object detection*" and "*open a new direction in this research domain*".
>
> Existing unsupervised object-discovery approaches [6,7] leverage the attention map of DINO [8] to perform Normalized Cut for pseudo-box generation. Notably, the generated pseudo-boxes are devoid of category annotations. As a result, existing unsupervised detectors fail to acquire any category-aware information. Furthermore, neither unsupervised nor one-shot detection methods [1,2,3] can learn category-aware features in the absence of category labels.
>
>
> (2) **We propose a feature similarity (FS) loss, which is specifically designed for reference-based unsupervised object detection training without categorey labels.** Conventional loss functions cannot train classification tasks without category labels. Our approach is non-trivial. We replace category confidence with feature similarity and use FS loss to enable category-aware learning without category labels. As shown in Table 3 of the main paper, FS loss is critical for object classification and achieves strong performance in both category-agnostic and category-aware detection tasks.
>
> (3) **We propose an unsupervised reference-based open-set object detection network, which is more concise and efficient.** It requires no manual annotations for training. RefCD encodes reference images and target images to be detected only through a reference encoder. Previous supervised one-shot detection methods [2,3] rely on complex feature interactions to extract reference image information. Experimental results in Table 1 and Table 2 of the main paper further demonstrate the effectiveness of RefCD.

---

> ### Author Response · Authors · 2025-11-20
> **Authors' Rebuttal 2**
>
> **W3: Specific differences from existing methods**
>
> >Specifically, there already exists a line of work exploring exemplar-based or reference-guided detection and recognition across various domains—including detection, counting, and segmentation. In the detection community specifically, methods such as [1-3] have also utilized exemplar or reference inputs to guide detection, though typically in supervised or semi-supervised settings. In parallel, several text-driven approaches, for example [4-5], conditioning detectors on textual prompts rather than visual exemplars.
>
> Thank you for sharing these related works. We have added these citations and conducted discussions in the main paper.
> We provide a detailed discussion between the related works you mentioned and RefCD:
>
> (1) LOD [1] is a non deep learning based object detection method. It extracts features by the pre-trained VGG16[9] and employs a graph ranking algorithm for object detection. Subsequently, the k-means algorithm is used to cluster the objects detected in each image. RefCD is a deep learning based unsupervised object detection method. It employs reference images to identify objects of interest and training without any manual annotations. Compared with RefCD, LOD aims to detect all objects in images rather than specific objects of interest categories. Moreover, k-means cannot correctly classify objects into categories.
>
> (2) Although Os2d [2] and Siamese-DETR [3] also utilize exemplars or reference inputs to guide detection, they require large-scale manually annotated data, especially category annotations, for supervised training. In contrast, RefCD requires no manual annotations, significantly reducing training costs. Notably, we also propose the feature similarity (FS) loss for unsupervised training, while other detectors [2,3,4,5] cannot learn category information in the absence of category annotations.
>
> (3) OWL-ViT [4] and Regionclip [5] are open-vocabulary detection methods. They utilize text descriptions as prompts for one-shot detection. This requires pre-trained large-scale visual-language model (VLM) (*e.g.*, CLIP [10]) as text encoder. These large-scale VLMs demand over 400 million image-text pairs for pre-training, which need expensive annotation costs. In contrast, RefCD only uses 1.5 million unannotated images for unsupervised training, which does not require any labor cost. In addition, OWL-ViT and Regionclip need to fine tune the encoder during training, which significantly increases the training cost.
>
> In summary, existing unsupervised object discovery methods cannot perform category-aware detection and existing detection frameworks require large amounts of manually annotated data. All of them cannot learn category information in the absence of category labels.
>
> ### R-Table 1: Different with existing methods.
>
> | Method | Deep learning based | Paradigm | Train with category label | large-scale VLM |
> | :----- | :-----------------: | :------: | :-----------------------: | :-------------: |
> | LOD [1] | × | Unsupervised | × | × |
> | Os2d [2] | √ | Supervised | √ | × |
> | Siamese-DETR [3] | √ | Supervised | √ | × |
> | OWL-ViT [4] | √ | Supervised | √ | √ |
> | Regionclip-ViT [5] | √ | Supervised | √ | √ |
> | RefCD(Ours) | √ | Unsupervised | ×  | × |
>
> [1] Vo, Van Huy, et al. "Large-scale unsupervised object discovery." Advances in Neural Information Processing Systems 34 (2021): 16764-16778.
> [2] Osokin, Anton, Denis Sumin, and Vasily Lomakin. "Os2d: One-stage one-shot object detection by matching anchor features." European Conference on Computer Vision. Cham: Springer International Publishing, 2020.
> [3] Liu, Qiankun, et al. "Siamese-DETR for generic multi-object tracking." IEEE Transactions on Image Processing 33 (2024): 3935-3949.
> [4] Minderer, Matthias, et al. "Simple open-vocabulary object detection." European conference on computer vision. Cham: Springer Nature Switzerland, 2022.
> [5] Zhong, Yiwu, et al. "Regionclip: Region-based language-image pretraining." Proceedings of the IEEE/CVF conference on computer vision and pattern recognition. 2022.
> [6] Wang, Yangtao, et al. "Tokencut: Segmenting objects in images and videos with self-supervised transformer and normalized cut." IEEE transactions on pattern analysis and machine intelligence 45.12 (2023): 15790-15801.
> [7] Wang, Xudong, et al. "Cut and learn for unsupervised object detection and instance segmentation." Proceedings of the IEEE/CVF conference on computer vision and pattern recognition. 2023.
> [8] Caron, Mathilde, et al. "Emerging properties in self-supervised vision transformers." Proceedings of the IEEE/CVF international conference on computer vision. 2021.
> [9] Simonyan, Karen, and Andrew Zisserman. "Very deep convolutional networks for large-scale image recognition." arXiv preprint arXiv:1409.1556 (2014).
> [10] Radford, Alec, et al. "Learning transferable visual models from natural language supervision." International conference on machine learning. PmLR, 2021.

---

> > ### Comment · Reviewer_eSLo · 2025-11-21
> >
> > Thank you for your response. I appreciate the clarifications and the additional visualizations provided. After carefully reading your responses, I would like to further explain the concerns behind my original questions.
> >
> > My question was aligned with Reviewer DSqC’s concern: "how the method handles significant intra-class appearance differences". Based on your current explanation and the examples in the manuscript, it seems that the model’s ability to generalize across intra-class variants is largely dependent on manually choosing a similarity threshold. The manuscript sets this threshold to 0 and achieve the best performance. While this may work for broad categories such as “dog” or “apple,” it also raises a new question: when visually similar but semantically different objects co-occur. For example, an unusually colored “orange apple” and an actual “orange”, could this lead to noticeable category drift?
> >
> > You emphasize that the proposed method is unsupervised and distinct from VLM-based approaches mainly due to differences in using pre-training data and its scale. However, the current manuscript relies on DINOv2 (ViT-Large), and other pretrained ResNet backbone. From a methodological standpoint, this still constitutes heavy reliance on large-scale pretrained visual models. Therefore, framing the distinction from CLIP and similar VLMs purely based on the availability of text-image pairs or annotation cost feels somewhat incomplete.
> >
> > Again, thank you for your detailed rebuttal. I hope these clarifications help explain my concerns.

---

> > > ### Author Response · Authors · 2025-11-24
> > >
> > > **Dear Reviewer eSLo,**
> > >
> > > **Thank you for taking the time to provide additional feedback and share your concerns with us.**
> > > **We sincerely appreciate your comments and would like to address your points with the following clarifications.**
> > >
> > > >My question was aligned with Reviewer DSqC’s concern: "how the method handles significant intra-class appearance differences". Based on your current explanation and the examples in the manuscript, it seems that the model’s ability to generalize across intra-class variants is largely dependent on manually choosing a similarity threshold. The manuscript sets this threshold to 0 and achieve the best performance. While this may work for broad categories such as “dog” or “apple,” it also raises a new question: when visually similar but semantically different objects co-occur. For example, an unusually colored “orange apple” and an actual “orange”, could this lead to noticeable category drift?
> > >
> > > Thank you for the insightful comments.
> > >
> > > Based on manually set similarity thresholds, RefCD shows robustness to intra-class variants, as illustrated in Figure 4 of the main paper.
> > > However, in addition to intra-class variations, visually similar but semantically distinct objects present another challenge for existing detectors.
> > > A similar limitation exists for both supervised and unsupervised detectors.
> > > To further verify the generalization ability of RefCD, we report the qualitative experimental results of DETR [1], Siamese-DETR [2], and RefCD on relevant cases (*i.e.*, orange apples and oranges) in Figure 13 of main paper.
> > > As shown in Figure 13(a) and Figure 13(b), RefCD can distinguish between visually similar but semantically distinct objects without causing noticeable category drift, while supervised-trained DETR cannot distinguish between orange apples and oranges.
> > >
> > > However, the ability of RefCD to handle such cases is limited. As illustrated in Figure 13(c) and Figure 13(d), although RefCD and Siamese-DETR can distinguish between orange apples and oranges to a certain extent, both orange apples and oranges exhibit almost the same similarity (with a difference of about 0.1) to the reference image (*i.e.*, either orange apple or orange).
> > > Matched objects typically yield higher feature similarity, enabling distinction between them using more refined similarity thresholds. (More details are in Appendix A.10)
> > >
> > > [1] Carion, Nicolas, et al. "End-to-end object detection with transformers." European conference on computer vision. Cham: Springer International Publishing, 2020.
> > > [2] Liu, Qiankun, et al. "Siamese-DETR for generic multi-object tracking." IEEE Transactions on Image Processing 33 (2024): 3935-3949.
> > >
> > > >You emphasize that the proposed method is unsupervised and distinct from VLM-based approaches mainly due to differences in using pre-training data and its scale. However, the current manuscript relies on DINOv2 (ViT-Large), and other pretrained ResNet backbone. From a methodological standpoint, this still constitutes heavy reliance on large-scale pretrained visual models. Therefore, framing the distinction from CLIP and similar VLMs purely based on the availability of text-image pairs or annotation cost feels somewhat incomplete.
> > >
> > > We deeply apologize for the confusion.
> > >
> > > We commit that we do require pre-trained DINOv2/ResNet as the reference encoder/backbone.
> > > However, the DINOv2 and ResNet used in RefCD are self-supervised/unsupervised pre-trained, without requiring manual annotations.
> > > We acknowledge that supervised pre-trained models can also work, but to reduce reliance on annotated data, we exclusively adopt unsupervised pre-trained models.
> > >
> > > However, we would like to clarify that another distinction between RefCD and open-vocabulary methods lies in the task settings. We leverage reference images as the prompt for detection, and only a single modality is involved in the task setting, which makes the unsupervised object detection possible since there exist lots of works on self-/unsupervised representation learning methods/models.
> > > This is also the reason we use images as references (*i.e.*, retaining only one modality).
> > > For VLM-based methods, two modalities are involved (*i.e.*, text and image), which requires feature alignment between text and images.
> > > To date, there is no unsupervised cross-modal pre-training method capable of aligning text and images, and pre-training requires the expensive manual annotations.

---

> > > > ### Comment · Reviewer_eSLo · 2025-11-25
> > > >
> > > > Thank you for the extra clarifications. The new explanations and comparisons do make the paper clearer. Some of my earlier concerns are still there, but I now better understand your choices and how you position the work.
> > > >
> > > > Given the overall structure and the solid experiments, I will keep your responses in mind when updating my score. My remaining concerns are mostly about the limitations of the method itself, but they do not detract from the good presentation and the amount of work you have put in.

---

> > > > > ### Author Response · Authors · 2025-11-25
> > > > >
> > > > > Dear reviewer eSLo,
> > > > >
> > > > > Your thoughtful consideration of our responses is sincerely appreciated. Engaging in this discussion with you has been truly rewarding.
> > > > > We welcome any further questions or discussions and appreciate the opportunity to improve our work.
> > > > >
> > > > > Thank you once again for your valuable time and effort!
> > > > >
> > > > > Best regards,
> > > > >
> > > > > The Authors

---

### Author Response · Authors · 2025-11-20

Dear Reviewers,

We sincerely thank you for your thoughtful and constructive feedback. We have dedicated considerable time to crafting this rebuttal, aiming to  explain the methodology of our work and supplement experimental results. The specific areas addressed include:

1. Differences with previous works (Reviewer **eSLo**, **DSqC**, **4kU3**)
2. Intra-class variation (Reviewer **eSLo**)
3. Partial object as reference (Reviewer **DSqC**)
4. More detailed experiments (Reviewer **VNJE**)
5. Closely related works (Reviewer **4kU3**)


We have updated our paper with the revisions and additional experimental results in the revised PDF. We will respond to each reviewer's comments individually over the next couple of days.

Once again, we are grateful for your valuable insights, which have significantly enhanced our work. We have made every effort to comprehensively address all concerns.

Thank you for your time and consideration.

Sincerely,

The Authors

---

### Author Response · Authors · 2025-12-01
**Summary of Rebuttal for AC**

**Dear Area Chair,**

**We sincerely appreciate the valuable time you have dedicated to reviewing our work. We hereby briefly summarize the rebuttal.**

(1) Under the circumstances of information leakage, we solemnly commit that **we have strictly adhered to academic norms**, and we have not, do not, and will not attempt to abuse the leaked information or make a connection with reviewers. If we ever did so, we are willing to accept reports from anyone and bear the corresponding penalties.

(2) We would like to express our gratitude to all reviewers for their recognition of our work, including:
**"experimental analysis is good and extensive"** (Reviewer eSLo);
**"a novel perspective on unsupervised object detection" and "opening a new direction in this research area"** (Reviewer DSqC);
**"a concise and well-structured pipeline", "the experimental evaluation is thorough", and "suggests a possible path toward reducing annotation costs in Open Vocabulary Object Detection"** (Reviewer VNJE);
and **"interesting and fundamental", "the idea of building amodal segmentor based on SAM is reasonable and make sense"** (Reviewer 4kU3).

(3) During the rebuttal phase, we have provided detailed responses to all reviewers' comments and concerns. Three of the four reviewers (except Reviewer VNJE) have further discussed with us, and we believe that all the concerns raised by all reviewers have been **adequately addressed**. Specifically:

- The last comment of **Reviewer eSLo** is: "Given the **overall structure and the solid experiments**, I will keep your responses in mind when **updating my score**. My remaining concerns are mostly about the limitations of the method itself, but **they do not detract from the good presentation and the amount of work you have put in**."

- The last comment of **Reviewer 4kU3** is: "After the rebuttal, my concerns are well addressed. Overall, **the task is practical, and the method is adequate.** Therefore, **I will update my ratings if feasible** (the edit button is invisible at this time)."

- The last comment of **Reviewer DSqC** is about his/her **misunderstanding of the category definition in our work**. And we have clarified in our response and and the paper(Line 196-198) that our work does not define categories since **our work is a fully unsupervised method without using category labels**.

- As for **Reviewer VNJE**, we believe that all his/her concerns have been addressed, as we followed up twice invitation to continue the discussion if he/she had any other concerns, but **the reviewer did not raise any additional concerns**.

(4) The submitted paper has been updated to address all the concerns/advice from all reviewers. As **some reviewers explicitly indicated their intention to update their ratings, we sincerely hope that the AC carefully reviews our work and rebuttal to make a fair and impartial decision.**

Best regards,

The Authors

---

### Meta-Review · Area_Chair_Wpj9 · 2026-01-09

**Summary:**

This paper proposes Reference-based Category Discovery (RefCD), an unsupervised reference-based object detection framework that aims to enable category-aware detection without manual category labels. The method locate all objects via category-agnostic detection and then identify specific category objects by measuring feature similarity between predicted boxes and unlabeled reference images. All four reviewers initially gave the score 4 to the paper. After rebuttal, Reviewer 4kU3 explicitly indicates willingness to increase the score, while Reviewer eSLo notes that key limitations remain but acknowledges the paper is substantially clearer.

Despite these improvements, several main concerns remain insufficiently resolved.

A central unresolved issue is the conceptual definition of “category” in RefCD. Reviewer DSqC repeatedly questioned how categories are defined, organized, and differentiated when no category labels or ontology is used. While the authors clarify that RefCD does not explicitly define or predict categories and that COCO categories are used only for evaluation, this effectively redefines a “category” as whatever is most similar to a given reference image. This makes category discovery implicitly reference-conditioned and threshold-dependent, rather than a well-defined semantic grouping. As a result, RefCD is conceptually much closer to reference-conditioned similarity matching or exemplar-based detection than to true category discovery, leaving its distinction from one-shot or reference-based detection paradigms still unclear.

Concerns also remain regarding the strength and generality of the experimental validation. Reviewer VNJE pointed out that the method heavily relies on the semantic richness of DINOv2 features. Although the authors present results on industrial and underwater datasets, these are reported only for RefCD without comparative baselines, making it impossible to assess whether the observed performance reflects a genuine advantage of the proposed framework or simply the transferability of the pretrained backbone in simpler domains.

The single-object tracking (SOT) extension is also not convincingly supported. The rebuttal compares RefCD mainly against very old or weak baselines (e.g., KCF(2014), USOT(2021)), and RefCD performs only marginally better than KCF while remaining far behind stronger modern unsupervised trackers, such as ULAST (Unsupervised learning of accurate siamese tracking. CVPR 2022), or Diff-Tracker (Diff-Tracker: Text-to-Image Diffusion Models are Unsupervised Trackers, ECCV2024). As a result, the evidence provided does not support the claim that RefCD is a competitive or practically meaningful tracker, and the tracking results appear more like a proof-of-concept rather than a substantiated extension.

Therefore, the AC recommends rejection at this time.

**Reviewer Concerns:**

Addressed: 4kU3
partially addressed: eSLo
Unresolved: DSqC, VNJE

**Reviewer Scores:**

4kU3: 4->6
eSLo: 4, or 4->6
DSqC, VNJE: 4

---

### Decision · Program_Chairs · 2026-01-26

Reject